# MODEL-AGNOSTIC MEASURE OF GENERALIZATION DIFFICULTY

## ABSTRACT

The measure of a machine learning algorithm is the difficulty of the tasks it can perform, and sufficiently difficult tasks are critical drivers of strong machine learning models. However, quantifying the generalization difficulty of machine learning benchmarks has remained challenging. We propose what is to our knowledge the first model-agnostic measure of the inherent generalization difficulty of tasks. Our *inductive bias complexity* measure quantifies the total information required to generalize well on a task minus the information provided by the data. It does so by measuring the fractional volume occupied by hypotheses that generalize on a task given that they fit the training data. It scales exponentially with the intrinsic dimensionality of the space over which the model must generalize but only polynomially in resolution per dimension, showing that tasks which require generalizing over many dimensions are drastically more difficult than tasks involving more detail in fewer dimensions. Our measure can be applied to compute and compare supervised learning, reinforcement learning and meta-learning generalization difficulties against each other. We show that applied empirically, it formally quantifies intuitively expected trends, e.g. that in terms of required inductive bias, MNIST < CIFAR10 < Imagenet and fully observable Markov decision processes (MDPs) < partially observable MDPs. Further, we show that classification of complex images < few-shot meta-learning with simple images. Our measure provides a quantitative metric to guide the construction of more complex tasks requiring greater inductive bias, and thereby encourages the development of more sophisticated architectures and learning algorithms with more powerful generalization capabilities.

## 1 INTRODUCTION

Researchers have proposed many benchmarks to train machine learning models and test their generalization abilities, from ImageNet (Krizhevsky et al., 2012) for image recognition to Atari games Bellemare et al. (2013) for reinforcement learning (RL). More complex benchmarks promote the development of more sophisticated learning algorithms and architectures that can generalize better.

However, we lack rigorous and quantitative measures of the generalization difficulty of these benchmarks. Generalizing on a task requires both *training data* and a model's *inductive biases*, which are any constraints on a model class enabling generalization. Inductive biases can be provided by a *model designer*, including the choice of architecture, learning rule or hyperparameters defining a model class. While prior work has quantified the training data needed to generalize on a task (*sample complexity*), analysis of the required inductive biases has been limited. Indeed, the concept of inductive bias itself has not been rigorously and quantitatively defined in general learning settings.

In this paper, we develop a novel information-theoretic framework to measure a task's *inductive bias complexity*, the information content of the inductive biases. Just as sample complexity is a property inherent to a model class (without reference to a specific training set), inductive bias complexity is a property inherent to a training set (without reference to a specific model class). To our knowledge, our measure is the first quantification of inductive bias complexity. As we will describe, our measure quantifies the fraction of the entire hypothesis space that is consistent with inductive biases for a task given that hypotheses interpolate the training data; see Fig 1 for an illustration and Definition 1 for a formal definition. We use this inductive bias complexity measure as a measure of the generalization difficulty; we hope our measure can guide the development of tasks requiring greater inductive bias.

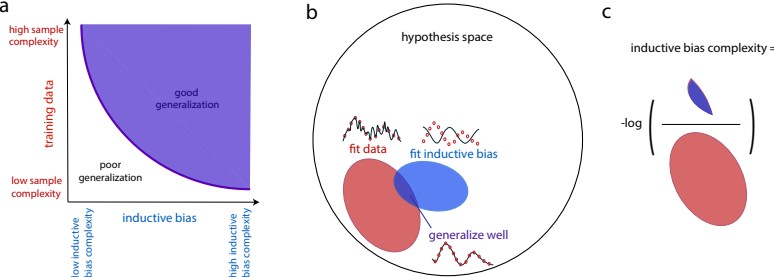

Figure 1: Dividing the hypothesis space of a task. (a) plots the trade-off between the amount of training data required (sample complexity) vs. the amount of inductive bias required (inductive bias complexity) to generalize well. Achieving a lower sample complexity for a fixed generalization performance target requires higher inductive bias complexity to further restrict the hypothesis space. While many papers have quantified sample complexity, we quantify inductive bias complexity. (b) categorizes hypotheses by whether or not they fit the training data or contain particular inductive biases. Both training data and inductive biases are required to generalize well. (c) shows how inductive bias complexity is computed: we quantify inductive bias complexity as negative log of the fraction of hypotheses that generalize well among the hypotheses that fit the training data.

As one concrete, but widely-applicable, suggestion, we find that adding Gaussian noise to the inputs of a task can dramatically increase its required inductive bias. We summarize our contributions as [1]:

- We propose a feature-based task framework to analyze generalization difficulty in various learning problems, including supervised learning, RL and meta-learning.

- We provide a formal, information-theoretic definition of *inductive bias complexity*.

- We develop a novel and to our knowledge first quantifiable measure of inductive bias complexity, which we propose using as a measure of the generalization difficulty of a task. This measure also allows us to quantify the relative inductive biases of different model architectures applied to a task.

- We propose a practical algorithm to estimate and compare the inductive bias complexities of tasks across the domains of supervised learning, RL and few-shot meta-learning. Empirically, we find that 1) partially observed RL environments require much greater inductive bias compared to fully observed ones and 2) simple few-shot meta-learning tasks can require much greater inductive bias than complex, realistic supervised learning tasks.

## 2 RELATED WORK

### 2.1 MODEL ORIENTED GENERALIZATION DIFFICULTY

The generalizability of machine learning models is traditionally quantified with *learning curves*, which relate the generalization error on a task to amount of training data (i.e. sample complexity) (Cortes et al., 1994; Murata et al., 1992; Amari, 1993; Hestness et al., 2017). Generalization error can be bounded by model class capacity measures, including Rademacher complexity (Koltchinskii & Panchenko, 2000) and VC dimension (Blumer et al., 1989), with simpler model classes achieving lower generalization error. While valuable, these measures often provide only loose, non-specific generalization bounds on particular tasks because they typically do not leverage a task's structure.

More recently proposed data-dependent generalization bounds yield tighter bounds on generalization error (Negrea et al., 2019; Raginsky et al., 2016; Kawaguchi et al., 2022; Lei et al., 2015; Jiang et al., 2021). These measure leverage the properties of a dataset to show that particular model classes will generalize well on the dataset. Relatedly, recent work on neural network scaling laws (Bahri et al., 2021; Hutter, 2021; Sharma & Kaplan, 2022) has modeled learning as kernel regression to show

---

[1]An anonymized version of our code can be found at: `https://anonymous.4open.science/r/task-difficulty-CF26`

among other results that sample complexity scales *exponentially* with the intrinsic dimensionality of data. Our approach is aligned in spirit to this line of work; however, instead of leveraging the structure of a task to bound generalization error or sample complexity, we instead bound the *inductive bias complexity* of a task; see Figure 1. We believe inductive bias complexity is a more appropriate measure of a task's difficulty compared to sample complexity. Notably, sample complexity assumes a specific model class; it is not model-agnostic. Thus, sample complexity only quantifies the contribution of training data towards solving a task given a particular model class; it does *not* quantify the difficulty of choosing an appropriate model class (e.g. a model architecture).

## 2.2 INFORMATION-THEORETIC MEASURES OF GENERALIZATION DIFFICULTY

Recent works have analyzed the properties of neural networks using information theory; for instance, information bottleneck can be used to extract well generalizing data representations (Shwartz-Ziv & Tishby, 2017; Saxe et al., 2019; Shamir et al., 2008). DIME (Zhang et al., 2020) uses information theory to estimate task difficulty, bounding the best case error rate achievable by *any* model on a dataset given the underlying data distribution. Achille et al. (2021) defines task difficulty based on the Kolmogorov complexity required to model the relationship between inputs and outputs. Although these measures can be related to generalization, they do not directly quantify the information needed to generalize. Alternatively, generalization difficulty can be expressed as the amount of information required to perform a task in addition to any training data provided to a learning system (Chollet, 2019). This aligns with other literature assessing the *prior knowledge* contained in a learning system whether through specific abilities (Hernández-Orallo, 2016), biases (Haussler, 1988), or model architectures (Du et al., 2018; Li et al., 2021). In this work, we take a similar approach, defining generalization difficulty as the amount of information in inductive biases required on top of training data in order to solve a task within the hypothesis space.

## 3 FEATURE-BASED TASK FRAMEWORK

It is well known that generalization from limited data can be viewed in a mathematically unified way for many learning problems including regression, classification and unsupervised learning (Vapnik, 1999). To analyze the generalization difficulty of many learning problems using common notation, we provide a unified notational framework that explicitly reveals the dimensions along which a learning system needs to generalize. Tab 1 summarizes our notation.

### 3.1 SETUP AND NOTATION

We define a task as consisting of *instances* $x$ and *actions* $y$; a learning system aims to learn a mapping from instances to actions. We assume that each instance $x$ is described by a set of *features*: $x$ has $K_c$ continuous features $c_1, c_2, ...c_{K_c}$ and $K_d$ discrete features $d_1, d_2, ...d_{K_d}$. Finally, we assume that the

Table 1: Table of symbols.

| Symbol | Meaning |
|---|---|
| $f(\cdot; \theta)$ | Function from the hypothesis space, parameterized by $\theta$ |
| $p$ | Input distribution, denotes the test distribution |
| $q$ | Input distribution, denotes the training distribution |
| $e(\theta, p)$ | Error of $f(\cdot; \theta)$ on input distribution $p$ |
| $\mathcal{L}$ | Loss function |
| $\mathcal{T}$ | Event that the hypothesis fits the training dataset |
| $\mathcal{K}$ | Event that the hypothesis fits inductive biases |
| $\varepsilon$ | Desired test error |
| $\bar{I}$ | Inductive bias complexity |
| $\Theta_q$ | Set of hypotheses that fit the training dataset |
| $L_\mathcal{L}$ | Lipschitz constant for $\mathcal{L}$ |
| $L_f$ | Constant used in second-order condition on $f$ |
| $m$ | Intrinsic dimensionality of the data |
| $k$ | Number of submanifolds |
| $r$ | Radius of the sphere on which datapoints lie |
| $d$ | Dimensionality of the model output $f(x; \theta)$ |
| $f_i$ | A single component of $f$ |
| $b$ | Upper bound on $\|f(x; \theta)\|$ |
| $n$ | Size of training dataset |
| $M$ | Maximum frequency of eigenfunctions considered |
| $\delta$ | Spatial resolution of $f$; equal to $2\pi r / M$ |
| $K$ | Maximum integer satisfying $\sqrt{K(K + m - 1)} \leq M$ |
| $E$ | # of eigenfunctions with frequency $\leq M$; $O(M^m)$ |

features uniquely describe all possible points $x$: there exists an invertible function $A$ such that:

$$x = A(c_1, c_2, ...c_{K_c}, d_1, d_2, ...d_{K_d}) \tag{1}$$

for all possible values of $x$ and the features on their respective manifolds. Note that such a decomposition of $x$ always exists by using a single feature $c_1 = x$ or $d_1 = x$; tasks with a more complex structure may have instances with a further decomposition reflecting the task structure.

We assume that for each instance $x$, there is a unique desired action $y \in \mathcal{Y}$ among the set of possible actions $\mathcal{Y}$; the optimal mapping is denoted by $y = f^*(x)$. In general, learning the exact mapping $f^*$ may be intractable. Instead, it may be feasible to closely match the function $f^*$ within some error. Given a loss function $\mathcal{L}$, we define the error of a function $f$ under a particular distribution $p$ over $x$ as:

$$\hat{e}(f, p) = \mathbb{E}_p[\mathcal{L}(f(x), x)] \tag{2}$$

### 3.2 Mapping the framework to diverse learning settings

Although the task framework above used notation that resembled supervised learning, our framework incorporates many different learning scenarios including unsupervised, reinforcement and meta learning. We summarize the notational mapping between our framework and various learning scenarios in Tab 6. Further details of this mapping are given in App A.

Our framework reveals the dimensions along which a task requires generalization. We use this in Section 4 where we quantify the difficulty of *generalizing* on a task. However, our framework may conceal aspects of a task relevant to other notions of difficulty. For instance, in online RL, we do not capture the policy-dependence of the training data; thus, we cannot quantify the exploration-exploitation trade-off. However, our framework captures exactly the *generalization-relevant* aspects of a task: in RL, for instance, generalization can be viewed in much the same way as supervised learning, except with different policies inducing different distributions during training and testing (Kakade, 2003). Importantly, in our analysis, we do not require that training and testing must be done on the same distribution (see Section 4.2); thus, we believe our framework applies to studying generalization in situations like online RL and beyond where the training and test distributions may be quite different.

## 4 Measuring Information Content of Inductive Biases

It is well known by the No Free Lunch theorem (Wolpert, 1996) that no learning algorithm can perform well on *any* task; only learning algorithms that possess appropriate inductive biases relevant to a particular task(s) can be expected to generalize well given a finite amount of data. Thus, generalizing on a task requires both training data and inductive biases (i.e. a choice of model class) (see Fig 1). In Section 1, we formally define *inductive bias complexity* as the amount of inductive bias required to generalize. In Section 4.2, we exactly bound this quantity in terms of the desired error rate of a task and distance between training and test distributions. In Section 4.3, we further approximate the bound to produce a practically computable estimate.

### 4.1 Formally Defining Inductive Bias Complexity

We formally relate inductive biases and generalization using the concept of a *hypothesis space*. Hypotheses may or may not satisfy a training set or a set of inductive biases (see Fig 1 (b)); generalization fundamentally requires both inductive biases and training data to specify a set of well-generalizing hypotheses. Note that there is a trade-off between the amount of training data and inductive bias required to generalize (see Fig 1 (a)). We aim to quantify the level of inductive bias required for generalization given a fixed amount of training data. We call this measure the *inductive bias complexity* (see Fig 1 (c)). Just as sample complexity quantifies the amount of training data required to generalize given a fixed inductive bias (i.e. a fixed model class), inductive bias complexity measures the amount of inductive bias required given a fixed amount of training data.

We begin by defining a (very broad and general) hypothesis space, consisting of functions $f(x; \theta)$ parameterized by a vector $\theta$. The hypothesis space is *not* the same as the model class (e.g. neural networks) used to solve a task. Instead, the hypothesis space is ideally a massive space that encompasses all possible model classes that might reasonably be used to solve a task. We view the hypothesis space constructed here simply as a mathematical tool to analyze the properties of the task itself rather than relating to the actual models trained on a task. We use a Bayesian framework, assuming that $f$ is drawn from some distribution and that the prior distribution of $\theta$ is uniform (which can generally be made to hold by reparameterizing). We assume that the true (target) function to be learned is expressible within this large hypothesis space, and we define it to be given by $f^*(x) = f(x; \theta^*)$. The generalization error of a specific $\theta$ and input distribution $p$ is denoted $e(\theta, p) = \hat{e}(f(\cdot; \theta), p)$. We aim to find a hypothesis $\theta$ achieving generalization error below a threshold $\varepsilon$: $e(\theta, p) \leq \varepsilon$. The training data and inductive biases both provide constraints on the hypothesis space; we wish to quantify how much the inductive biases need to constrain the space to yield a well-generalizing set of hypotheses.

Intuitively, the inductive bias required to achieve a particular error rate is related to *additional* information required by the inductive biases beyond any task-relevant information provided by training data. Specifically, inductive bias complexity can be viewed as the additional *information*

required to specify the well-generalizing hypotheses *given the set of hypotheses that fit the training data*. Thus, we formally define inductive bias complexity $\tilde{I}$ as follows:

**Definition 1.** *Suppose a probability distribution is defined over a set of hypotheses, and let random variable $\theta$ correspond to a sample from this distribution. Given an error threshold $\epsilon$ which models are optimized to reach on the training set, the inductive bias complexity required to achieve error rate $\varepsilon$ is on a task*

$$\tilde{I} = -\log \mathbb{P}(e(\theta, p) \leq \varepsilon \mid e(\theta, q) \leq \epsilon) \tag{3}$$

Observe that this is exactly the information content of the probabilistic event that a hypothesis that fits the training set also generalizes well. For ease of analysis, we define "fitting" the training set as perfectly interpolating it (setting $\epsilon = 0$). A well-generalizing hypothesis, achieving a low but non-zero error rate, might *not* interpolate the training set. Nevertheless, the regime of perfect interpolation can be relevant for training overparameterized deep networks; we leave other regimes as future work and briefly outline how to extend to these regimes in App G.

## 4.2 An Exact Upper Bound on Inductive Bias Complexity

We are able to provide an exact upper bound on the inductive bias complexity given above, under some technical assumptions. These assumptions may not hold in all settings; nevertheless, we believe they hold in many common settings (e.g. when the loss function is a square loss and the model $f$ is twice differentiable with bounded inputs and parameters); App C discusses the validity of these assumptions. Assume that the input, parameter, and output spaces are equipped with standard distance metrics $d(\cdot, \cdot)$ and norms $\| \cdot \|$. Let $p$ denote the test distribution, $q$ denote the training distribution, and $W(\cdot, \cdot)$ denote the 1-Wasserstein distance. Additionally define $\Theta_q = \{\theta \mid e(\theta, q) = 0\}$ to be the set of hypotheses that perfectly interpolate the training dataset.

**Theorem 1.** *Suppose that the loss function satisfies $\mathcal{L}(y, x) \geq 0$, and equality holds if and only if $y = f^*(x)$. Suppose that the loss function is invariant to shifts in $x$ in the sense that $\mathcal{L}(y + f^*(x_2) - f^*(x_1), x_2) = \mathcal{L}(y, x_1)$ for all $y, x_1, x_2$. We also $\mathcal{L}$ has a Lipschitz constant $L_{\mathcal{L}}$:*

$$\mathcal{L}(y + \delta, x) - \mathcal{L}(y, x) \leq L_{\mathcal{L}} \|\delta\| \tag{4}$$

*for all $y, \delta, x$. We also assume a second-order condition on $f$ for some constant $L_f$:*

$$\|f(x_1; \theta_1) - f(x_1; \theta_2) - f(x_2; \theta_1) + f(x_2; \theta_2)\| \leq L_f d(x_1, x_2) d(\theta_1, \theta_2) \tag{5}$$

*for all $x_1, x_2, \theta_1, \theta_2$. Then we have the following upper bound on the inductive bias complexity:*

$$\tilde{I} \leq -\log \mathbb{P}\left( d(\theta, \theta^*) \leq \frac{\varepsilon}{L_{\mathcal{L}} L_f W(p, q)} \mid \theta \in \Theta_q \right), \tag{6}$$

The bound depends on the following key quantities: desired error rate $\varepsilon$, interpolating hypothesis space $\Theta_q$ and 1-Wasserstein distance between the training and test distributions $W(p, q)$. See App B for a proof. Observe that the proof is quite general: it does not rely on any specific assumptions on the hypothesis space, nor does it assume any relationship between the test and training distributions.

## 4.3 Empirical Inductive Bias Complexity Approximation

We now apply a series of approximations on the general bound of Theorem 1 to achieve a practically computable estimate. We present this approximation in the setting of supervised classification, but note that it can be applied to other settings as we show in our experiments (see App D for full details on our empirical computation). As we propose the first measure of inductive bias complexity, our focus is on capturing general trends (scalings) rather than producing precise estimates. We hope future work may further refine these estimates or further relax our assumptions.

**Supervised Classification Setting**   We assume data points $x$ lie on $d$ non-overlapping $m$-spheres of radius $r$, where each sphere corresponds to a class. Furthermore, we assume that the test distribution $p$ is uniform over all the $m$-spheres and the training distribution $q$ is the empirical distribution of $n$ points drawn i.i.d. from $p$. This approximation captures key characteristics of data distributions in actual classification tasks: 1) bounded domain, 2) similar density over all points, 3) non-overlapping distributions for different classes. We assume the dimensionality of $f(x; \theta)$ is $d$, the number of

classes. Let $b$ be a constant such that $\|f(x;\theta)\| \leq b$ for all $x, \theta$. Let $\delta$ be the "spatial resolution" of $f$, defined such that perturbing the input $x$ by amounts less than $\delta$ does not significantly change the output $f(x;\theta)$ (i.e. $\delta$ is the minimum change that $f$ is sensitive to). In our case of supervised classification, $\delta$ corresponds to the distance between classes.

**Parametrization of Hypotheses**  In order to compute Equation 6, we must make a specific choice of parameterization for $f(x;\theta)$. Recall from Section 1 that we wish to select a broad and general hypothesis space encompassing all model classes practically applicable to a task. Hypotheses that we may *not* want to consider include functions which are sensitive to changes on $x$ below the task relevant resolution $\theta$. Thus, we consider only bandlimited functions with frequencies less than $M = 2\pi r/\delta$.

More formally, to construct a basis of functions below a certain frequency on a sphere, we use the approach in McRae et al. (2020), which parameterises functions as a linear combination of eigenfunctions of the *Laplace-Beltrami operator* on the manifold. The Laplace-Beltrami operator is a generalization of the Laplacian to general manifolds; eigenfunctions of this operator are analogous to Fourier basis functions in Euclidean space. See Fig 6 for a visualization in the Euclidean case.

For each nonnegative integer $k$, the Laplace-Beltrami operator on the $m$-sphere has $\binom{m+k}{m} - \binom{m+k-2}{m}$ eigenfunctions with frequency $\sqrt{k(k+m-1)}$ (Dai & Xu, 2013). Taking $K$ to be the maximum integer $k$ such that $\sqrt{k(k+m-1)} \leq M$, there are

$$E = \sum_{k=0}^{K} \binom{m+k}{m} - \binom{m+k-2}{m} = \binom{m+K-1}{m} + \binom{m+K}{m} \tag{7}$$

eigenfunctions with frequency at most $M$. Note that for large $M$, $E$ scales as $O(M^m)$ since $K$ scales as $M$. Since each coefficient has a real and imaginary part, the dimensionality of the parameterization of a single component $f_k$ is $2E$. There are $d$ components on each of $d$ submanifolds, so the dimensionality of the full parameterization $\theta$ is $2d^2E$. Each training point constrains the dimensionality of the possible values of $\theta$ by $d$. Therefore, the dimensionality of $\Theta_q$ is $2d^2E - nd$. Next, we obtain a bound on $\|\theta\|$. If the parameters of a component $f_k$ are $\theta_k$, Parseval's identity (Stein & Shakarchi, 2011) states that $\|\theta_k\|^2$ equals the average value of $\|f_k(x;\theta_k)\|^2$, which is bounded by $b^2$. Since there are $d^2$ components, we have $\|\theta\| \leq bd$.

Next, to approximate the volume of $\Theta_q$, we must make an assumption on the shape of $\Theta_q$. Given that we have a bound on $\theta$, we simply approximate $\Theta_q$ as a disk of dimensionality $2d^2E - nd$ and radius $bd$; note that the disk is embedded in a $2d^2E$ dimensional space. This disk approximation retains key properties of interpolating manifolds (low-dimensionality and boundedness) while being analytically tractable; see App C.3 for further discussion.

The set $\left\{ \theta \in \Theta_q \mid d(\theta, \theta^*) \leq \frac{\varepsilon}{L_{\mathcal{L}} L_f W(p,q)} \right\}$ is a disc with radius $\frac{\varepsilon}{L_{\mathcal{L}} L_f W(p,q)}$. Therefore,

$$\mathbb{P}\left( d(\theta, \theta^*) \leq \frac{\varepsilon}{L_{\mathcal{L}} L_f W(p,q)} \mid \theta \in \Theta_q \right) \approx \left( \frac{\frac{\varepsilon}{L_{\mathcal{L}} L_f W(p,q)}}{bd} \right)^{2d^2E - nd}. \tag{8}$$

To arrive at this approximation, we set the hypothesis space as a linear combination of eigenfunctions of the Laplace-Beltrami operator on the manifold, and then approximated volumes in the hypothesis space as balls. Since we do not make specific assumptions on the manifold of $x$ to construct our hypothesis space, we expect the overall scaling of our result to hold for general tasks.

**Approximating Wasserstein Distance**  We obtain a simple scaling result for the Wasserstein distance $W(p,q)$, the expected transport distance in the optimal transport between $p$ and $q$. In the optimal transport plan, each training point is associated with a local region of the sphere; the plan moves probability density between each point and the local region around each point. These regions must be equally sized, disjoint, and tile the entire sphere. Although the regions' shapes depend on the specific distribution $q$, if $q$ is near uniform, we expect these regions to be similarly shaped.

We approximate these regions as hypercubes because they are the simplest shape capable of tiling $m$-dimensional space. In order to fill the full area of the sphere, the hypercubes must have side length

$$s = \left( \frac{1}{n} \cdot \frac{2\pi^{(m+1)/2}}{\Gamma\left(\frac{m+1}{2}\right)} r^m \right)^{1/m} = rn^{-1/m} \left( \frac{2\pi^{(m+1)/2}}{\Gamma\left(\frac{m+1}{2}\right)} \right)^{1/m}. \tag{9}$$

The expected distance between two randomly chosen points in a hypercube is at most $s\sqrt{m/6}$ (Anderssen et al., 1976). We use this distance as an approximation of the optimal transport distance between each point and its associated local hypercube. The final $W(p,q)$ can then simply be approximated as the optimal transport distance corresponding to each point:

$$W(p,q) \approx s\sqrt{\frac{m}{6}} = rn^{-1/m}\sqrt{\frac{m}{6}}\left(\frac{2\pi^{(m+1)/2}}{\Gamma\left(\frac{m+1}{2}\right)}\right)^{1/m} \in O(rn^{-1/m}) \tag{10}$$

In App C.4, we empirically compute Wasserstein distances on a sphere and find that the distances follow this approximation. Although this particular approximation is valid when the empirical training distribution $q$ is drawn i.i.d. from a uniform underlying data distribution $p$, our framework can be applied to non-i.i.d. settings by simply modifying our Wasserstein distance approximation to reflect the train-test distributional shift.

**Final Approximation** Since $f$ is a linear combination of basis functions with coefficients $\theta$, $L_f$ upper bounds the norm of the gradient of the basis functions. A basis function with a single output and frequency $\sqrt{k(k+m-1)}$ has a gradient with magnitude at most $k/r$. Since $f$'s output has dimensionality $d$, we can take $L_f = K\sqrt{d}/r$. Substituting the result of Equation 8 along with our approximation for $W(p,q)$ into Equation 6 gives an approximated inductive bias complexity of

$$\tilde{I} \approx (2d^2E - nd)\left(\log b + \frac{3}{2}\log d + \log K - \frac{1}{m}\log n - \log\frac{\varepsilon}{L_{\mathcal{L}}} + \log c\right). \tag{11}$$

where $c = \sqrt{\frac{m}{6}}\left(\frac{2\pi^{(m+1)/2}}{\Gamma\left(\frac{m+1}{2}\right)}\right)^{1/m}$, which is roughly constant for large intrinsic dimensionality $m$. Recall that $d$ is the number of classes, $n$ is the number of training data points, $\varepsilon$ is the target error, $L_{\mathcal{L}}$ is the Lipschitz constant of the loss function, $b$ is a bound on the model output, and $K$ and $E$ are functions of $m$ and the maximum frequency $M = 2\pi r/\delta$.

**Properties of Inductive Bias Complexity** The dominant term in our expression, Equation 11, for inductive bias complexity is $2d^2E - nd$. Since $E \in O(M^m)$, inductive bias complexity is *exponential* in the data's intrinsic dimensionality $m$ (see App G for further intuition). Additionally, $M = 2\pi r/\delta$, so for a fixed $m$, inductive bias complexity is polynomial in $1/\delta$. Thus, $\delta$ impacts inductive bias complexity less than $m$, but can still be significant for large $m$. Inductive bias complexity increases roughly linearly in $Z$ and $D$, so these values have less impact than $m$ and $\delta$. Inductive bias complexity decreases roughly linearly with the number of training points $n$; intuitively, this is because each training point provides a roughly equal amount of information. Finally, inductive bias complexity increases logarithmically as the desired error $\varepsilon$ decreases. Intuitively, the inductive bias complexity depends on the desired performance level: achieving the performance of a random classifier on a classification task is trivial, for example.

## 5 EMPIRICAL GENERALIZATION DIFFICULTY OF BENCHMARKS

Here, we use our quantification of inductive bias complexity as a measure of the *generalization difficulty* of a task (or *task difficulty*). Inductive bias complexity corresponds to the effort that a model designer must apply to generalize on a task, while sample complexity corresponds to the contribution of the training data; thus, inductive bias complexity is an appropriate measure of the generalization difficulty of a task from a model designer's perspective. We empirically validate our measure on several supervised classification, meta-learning, and RL tasks. We also quantify the relative inductive biases of different model architectures on different tasks and empirically show task difficulty trends over parametric task variations.

Computing the difficulty $\tilde{I}$ of Equation 11 for specific tasks specifying several parameters. The target generalization error $\varepsilon$ is fixed at a level based on the type of the task (see App D Tab 3). We estimate the data dimensionality $m$ by relating the decay rate of nearest neighbor distances in the training set to intrinsic dimensionality (Pope et al., 2021). The required detail (resolution) per dimension $\delta$ is set to the inter-class margin for classification tasks and a scale at which state perturbations do not significantly affect trajectories for RL tasks. See App D for full details.

## 5.1 TASK DIFFICULTY OF STANDARD BENCHMARKS

**Image classification** We estimate the task difficulty of the commonly used image classification benchmarks MNIST, SVHN (Netzer et al., 2011), CIFAR10 (Krizhevsky et al., 2009), and ImageNet (Deng et al., 2009). As shown in Fig 2, models solving these tasks encode large amounts of inductive bias, with more complex tasks requiring models with more inductive bias.

**Few-shot learning** We estimate the task difficulty of 1-shot, 20-way classification on Omniglot (Lake et al., 2015). The model must learn a function $f$ mapping a *dataset* of 20 images from an alphabet to a *classifier* $g$, which itself maps an image to a probability distribution over 20 classes. Both the inputs and outputs of $f$ have much larger dimensionality than for typical image classification. This yields very large task difficulties: Omniglot has task difficulty $\approx 10^{145}$ bits (vs. $10^{41}$ for ImageNet); see Fig 2.

**RL** We find the task difficulty of the Reacher, Hopper, and Half-Cheetah control tasks in MuJoCo (Todorov et al., 2012; Duan et al., 2016). Fig 2 shows that more complex environments with higher dimensional observation and action spaces have higher task difficulties as intuitively expected. Moreover, the range of task difficulties is comparable with image classification: task difficulties are comparable across domains.

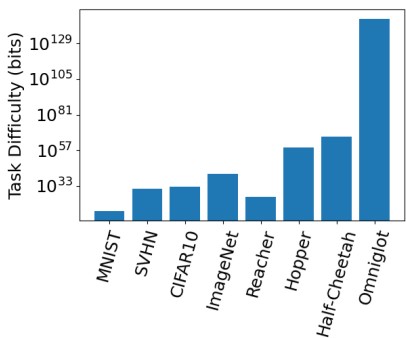

Figure 2: Task difficulties of various benchmark tasks across domains.

## 5.2 INDUCTIVE BIAS CONTRIBUTED BY DIFFERENT MODELS TO VARIOUS TASKS

Although our task difficulty measure is model-agnostic, we can use it to compare how much inductive bias different models contribute to various tasks. To do this, we find the test set performance of a trained model from the model class (e.g. test set accuracy for classification tasks). We then use this performance level to compute $\varepsilon$ in the task difficulty expression in Equation 11 (importantly, we do not *directly* use the test set data). This method is motivated by the idea that a model class enables generalization by providing an inductive bias, and its generalization ability can be quantified using test set performance. Thus, we quantify a model class's inductive bias as the minimum inductive bias required to achieve the performance of the model class: greater performance requires greater inductive bias. As observed in Tab 2 (see Tab 7 for full results), the increase in information content resulting from using a better model architecture (such as switching from ResNets (He et al., 2016) to Vision Transformers (Dosovitskiy et al., 2021)) is generally larger for more complex tasks. In other words, more complex tasks *extract* more of the inductive bias information present in a model.

Table 2: The inductive bias information contributed by different model architectures for image classification tasks. (FC-$N$ refers to a fully connected network with $N$ layers.)

| CIFAR10 | | ImageNet | |
|---|---|---|---|
| Model | Information Content ($\times 10^{32}$ bits) | Model | Information Content ($\times 10^{41}$ bits) |
| Linear (Nishimoto, 2018) | 2.250 | Linear (Karpathy, 2015) | 2.063 |
| AlexNet (Krizhevsky et al., 2012) | 2.709 | AlexNet (Krizhevsky et al., 2012) | 2.290 |
| ResNet-50 (Wightman et al., 2021) | 3.195 | ResNet-50 (He et al., 2016) | 2.362 |
| BiT-L (Kolesnikov et al., 2020) | 3.453 | BiT-L (Kolesnikov et al., 2020) | 2.449 |
| ViT-H/14 (Dosovitskiy et al., 2021) | 3.513 | ViT-H/14 (Dosovitskiy et al., 2021) | 2.461 |

## 5.3 TRENDS OF TASK DIFFICULTY

In this section, we consider how task difficulty changes as we parametrically vary aspects of a task. Please see App F for additional task variations.

**Varying training data** We vary the amount of training data provided in ImageNet and find in Fig 3a that the task difficulty decreases with more training points. Observe that even when training data is increased by many orders of magnitude, the scale of task difficulty remains large, suggesting that training data provides relatively little information to solve a task relative to inductive biases. Intuitively,

this is because in the absence of strong inductive biases, training data only provides information about the approximated function in a *local* neighborhood around each training point; inductive biases, by contrast, can provide *global* constraints on the function. See App G for further discussion.

**Varying desired error rate** In Fig 3b, we vary the desired error on the Reacher task and find that task difficulty decreases with desired error rate as expected. As with varying training data, the scale of task difficulty remains large even when dramatically reducing the desired error rate.

**Varying intrinsic dimension** We consider a noisy version of Cartpole (Florian, 2007), an inverted pendulum control task where agents apply binary forces to control a pole. In noisy Cartpole, the agent observes a noisy version of the pole's angular position and velocity, requiring possibly multiple observations to find the

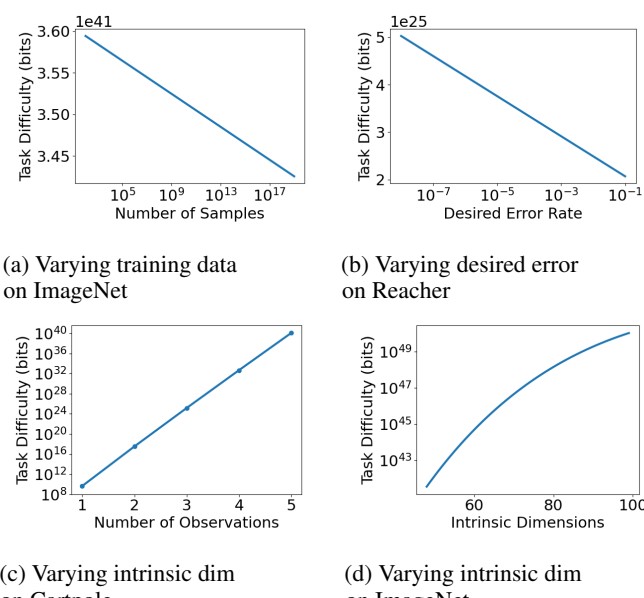

(a) Varying training data on ImageNet

(b) Varying desired error on Reacher

(c) Varying intrinsic dim on Cartpole

(d) Varying intrinsic dim on ImageNet

Figure 3: Task difficulties on variations of benchmark tasks.

optimal action. The agent must learn a function with a higher-dimensional input, making the task more difficult. As Fig 3c shows, task difficulty is exponential in the number of observations needed to find the optimal action. We also increase the intrinsic dimensionality of ImageNet classification by adding perturbations to the images outside the original image manifold; creating classifiers robust to such perturbations is known theoretically and empirically to be difficult (Madry et al., 2017; Yin et al., 2019). Fig 3d shows that task difficulty drastically increases with added intrinsic dimensions. Thus, increasing the intrinsic dimension of a task is a powerful way of increasing its difficulty.

## 6 DISCUSSION

For all tasks, inductive bias complexities are quite large, much larger than practical model sizes. This is due to the vast size of the hypothesis space constructed in Section 4.3; it includes any bandlimited function below a certain frequency threshold. Specifying a well-generalizing subset in such a massive hypothesis space requires massive amounts of information. See App G for further discussion.

Our measure reveals that tasks with inputs on higher dimensional manifolds require far more inductive bias relative to other variations of tasks. This yields counter-intuitive findings: meta-learning on simple Omniglot handwritten characters requires far more inductive bias than than classification of complex, realistic ImageNet images. These findings suggest that to construct tasks requiring high inductive bias, prioritizing generalization over many degrees of variation is more important than increasing the richness of the task within each dimension of variation. Good benchmarks would require generalization over high dimensional spaces, but, like Omniglot, may appear simple. Examples of such benchmarks include few-shot learning, adversarially robust image classification, and MDPs with very limited observations. Our measure provides a quantitative guide to designing such benchmarks. As a specific, but broadly-applicable example of increasing the number of dimensions of variation, we suggest adding Gaussian noise to task inputs.

To our knowledge, we provide the first measure of inductive bias complexity; thus, it is not designed to be quantitatively precise. We hope future work may be able to further refine our estimates. Nevertheless, we hope our measure may encourage the development of tasks requiring well-generalizing architectures and learning rules with many built-in inductive biases.

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

## A  SETTINGS OF THE FEATURE-BASED TASK FRAMEWORK

In this section, we provide detailed descriptions of the mapping of various learning settings to our framework.

**Supervised classification**  In a supervised classification setting, instances can be viewed as consisting of two features: a discrete feature $d_1$ corresponding to a class and a continuous feature $c_1$ specifying a particular example within the class:

$$x = A(c_1, d_1). \tag{12}$$

The optimal action function is to output the class $f^*(x) = d_1$. Observe that the continuous feature $c_1$ is irrelevant to the task; the goal of this task is to learn from training data how to extract the feature $d_1$ while maintaining invariance to $c_1$.

As a concrete example, consider the MNIST (Deng, 2012) classification dataset; here labels correspond to $d_1$, and $c_1$ corresponds to all other information necessary to specify each image (e.g. line width, rotation, overall size). In general, $c_1$ may not directly extractable from images; one method of estimating a representation of $c_1$ is to train a class-conditional autoencoder on MNIST images, and extract the latent representation of each image as $c_1$.

**Meta-learning for few-shot supervised classification**  In a few-shot meta-learning setting, instances can be viewed as consisting of two features: a continuous feature $c_1$ corresponding to a set of related classes, and a continuous feature $c_2$ corresponding to a set of samples from those classes.

$$x = A(c_1, c_2) \tag{13}$$

The objective is to output a function $g$ such that $g$ maps inputs in set of classes to corresponding labels: $f^*(x) = g$. Note that $g$ here depends only on $c_1$; the desired function $f^*$ is invariant to $c_2$. Typically, a meta-learning system will receive instances from a subset of $c_1$ values, and will be asked to generalize to a different subset within the same distribution. To allow the meta-learning system to associate instances with classes on the unseen $c_1$ values, a few inputs with the unseen $c_1$ are provided for each class.

An example of such a meta-learning problem is few-shot classification on the Omniglot (Lake et al., 2015) dataset. Here, $c_1$ corresponds to an alphabet, and $c_2$ corresponds to a set of sample images within that alphabet.

**Reinforcement learning**  In a reinforcement learning setting, instances can be viewed as consisting of a single feature specifying all the information in an instance. In the language of reinforcement learning, instances may typically correspond to the state of an environment for a fully-observed Markov decision process (MDP), or a sequence of observations for a partially-observed MDP. The single feature describing the instance may in general be discrete or continuous; for notational purposes, we default to continuous features when features may be either continuous or discrete.

$$x = A(c_1) \tag{14}$$

Here, $f^*(x)$ denotes the optimal action to take given state or observation sequence $c_1$. The goal of the learning system is to generalize over unseen states or observation sequences. However, note that the agent is not asked to generalize over parameters of the underlying MDP; the MDP may be fixed.

Note that our framework does not capture certain important aspects of RL settings; for instance, for online RL, it does not capture the fact that instances observed during may depend on the previous actions of the agent. This dependence is important to analyze the exploration-exploitation tradeoff of an RL task, for instance. Nevertheless, our framework is sufficient to assess the difficulty of generalizing in RL. The fact that the set of states or observation sequences over which is trained depends on the behavior of the agent has limited relevance to the generalization difficulty of the task.

As an example, consider the Cartpole continuous control task from the MuJoCo suite (Tassa et al., 2018). Here, $c_1$ corresponds to the observed state of the environment from which the agent is trained to extract the optimal action.

**Unsupervised autoencoding** In unsupervised autoencoding, the optimal action function $f^*(x)$ is simply the identity function $f^*(x) = x$. The goal is to generalize on a test set after observing a set of training instances. Usually, this is non-trivial as the learning system is constrained such that learning an identity function is infeasible. However, we highlight that our measure is model-agnostic; in this case the task difficulty can be viewed as a measure of complexity of the instance set. A specific example of such a problem may be unsupervised autoencoding of MNIST digits, each of which has a single continuous valued feature.

**Meta-reinforcement learning** In a meta-reinforcement learning setting, instances correspond to trajectories through *environments*, which consist of two features: a continuous feature $c_1$ parameterizing the environment and a continuous feature $c_2$ corresponding to the specific trajectory within the environment. Both features may in general be discrete or continuous; we use continuous features for our notation:

$$x = A(c_1, c_2) \tag{15}$$

The objective is to output a *policy* $g$ such that $g$ that maps states in an environment to actions: $f^*(x) = g$. Note that $g$ depends only on $c_1$; the desired function $f^*$ is invariant to $c_2$. The learning system is required to learn from the set of $c_1$ in the training set and generalize to an unseen set of $c_1$.

As an example, consider the Meta-World (Yu et al., 2019) meta-reinforcement learning benchmark in which an agent controlling a robotic arm must generalize from a set of training skills to an unseen set of test skills. The skills include simple manipulations such as pushing, turning, placing etc. In our formulation, $c_1$ corresponds to the skill and $c_2$ corresponds to specific trajectories of the arm performing the skill.

## B    PROOF OF THEOREM 1

*Proof.* Note that for any $x_1, x_2, \theta$, we have

$$\mathcal{L}(f(x_1; \theta), x_1) = \mathcal{L}\Big(f(x_2; \theta) + (f(x_1; \theta^*) - f(x_2; \theta^*))$$
$$+ (f(x_1; \theta) - f(x_1; \theta^*) - f(x_2; \theta) + f(x_2; \theta^*)), x_1\Big) \tag{16}$$

$$\leq \mathcal{L}\Big(f(x_2; \theta) + (f(x_1; \theta^*) - f(x_2; \theta^*)), x_1\Big)$$
$$+ L_{\mathcal{L}}\|f(x_1; \theta) - f(x_1; \theta^*) - f(x_2; \theta) + f(x_2; \theta^*)\| \tag{17}$$
$$\leq \mathcal{L}(f(x_2; \theta), x_2)$$
$$+ L_{\mathcal{L}}\|f(x_1; \theta) - f(x_1; \theta^*) - f(x_2; \theta) + f(x_2; \theta^*)\| \tag{18}$$
$$\leq \mathcal{L}(f(x_2; \theta), x_2) + L_{\mathcal{L}}L_f d(x_1, x_2)d(\theta, \theta^*), \tag{19}$$

where equation 17 is a result of the Lipschitz condition on $\mathcal{L}$, equation 18 is a result of the shift-invariance of $\mathcal{L}$ and equation 19 is a result of our condition on $f$. This allows us to bound the difference

$$e(\theta, p) - e(\theta, q) = \mathbb{E}_p[\mathcal{L}(f(x; \theta), x)] - \mathbb{E}_q[\mathcal{L}(f(x; \theta), x)] \tag{20}$$

as follows: let $\gamma$ be a distribution on $(x_1, x_2)$ with marginal distributions $p$ and $q$, respectively. Then

$$e(\theta, p) - e(\theta, q) = \mathbb{E}_\gamma[\mathcal{L}(f(x_1; \theta), x_1) - \mathcal{L}(f(x_2; \theta), x_2)] \tag{21}$$
$$\leq \mathbb{E}_\gamma[L_{\mathcal{L}}L_f d(x_1, x_2)d(\theta, \theta^*)]. \tag{22}$$

By the definition of Wasserstein distance,

$$e(\theta, p) - e(\theta, q) \leq L_{\mathcal{L}}L_f d(\theta, \theta^*) \inf_\gamma \mathbb{E}_\gamma[d(x_1, x_2)] = L_{\mathcal{L}}L_f d(\theta, \theta^*)W(p, q), \tag{23}$$

where the infimum is taken over all distributions $\gamma$ with marginal distributions $p, q$. Therefore, for all $\theta \in \Theta_q$ with $d(\theta, \theta^*) \leq \frac{\varepsilon}{L_{\mathcal{L}}L_f W(p,q)}$, we have $e(\theta, p) \leq e(\theta, q) + \varepsilon = \varepsilon$ (since $e(\theta, q) = 0$). Recall that we assume a uniform prior on $\theta$. Note that this can typically be made to hold with an appropriate choice of reparameterization: if our original density on $\theta$ is $P(\theta)$, then we may choose new parameters $\tilde{\theta} = g(\theta)$ satisfying $|det(\frac{\partial g(\theta)}{\partial \theta})| = P(\theta)$ to yield a uniform density over $\tilde{\theta}$.

Using the uniform prior assumption on $\theta$, this gives

$$\tilde{I} = -\log \mathbb{P}(e(\theta, p) \leq \varepsilon \mid e(\theta, q) = 0) \tag{24}$$

$$\leq -\log \mathbb{P}\left(d(\theta, \theta^*) \leq \frac{\varepsilon}{L_{\mathcal{L}} L_f W(p, q)} \mid \theta \in \Theta_q\right), \tag{25}$$

as desired. $\qquad\square$

## C  VALIDITY OF THEORETICAL ASSUMPTIONS

To arrive at our final expression for task difficulty in Equation 11, we make a number of assumptions. In this section, we further discuss the validity of each of our assumptions.

### C.1  SHIFT-INVARIANCE OF LOSS FUNCTION

In many supervised classification and regression settings, the shift invariance assumption is satisfied by a squared error loss function. For clarity, in supervised regression, we denote inputs as $x$ and outputs as $y$. A squared error loss function is constructed as $\mathcal{L}(y, x) = ||y - f^*(x)||_2^2$ which is shift invariant since

$$\mathcal{L}(y + f^*(x_1) - f^*(x_2), x_1) = ||y + f^*(x_1) - f^*(x_2) - f^*(x_1)||_2^2 = ||y - f^*(x_2)||_2^2 = \mathcal{L}(y, x_2). \tag{26}$$

Next, in a few-shot meta-learning setting, there are reasonably broad conditions under which shift-invariance holds: Instances correspond to datasets and actions correspond to functions mapping from inputs to the outputs of the inner loop task. For clarity, we define the inner task inputs and outputs as $i$ and $o$; the goal is to learn a mapping from datasets $x$ to functions $y$ that map $o = y(i)$. Furthermore, suppose a loss function $L(o, i)$ is defined for the inner loop task. Then, suppose the meta-loss function is defined as $\mathcal{L}(y, x) = \mathbb{E}_{i \sim x}[L(y(i), i)]$. We assume that functions $y$ are additive in the following sense: for any $y_1, y_2, x$, $(y_1 + y_2)(x) = y_1(x) + y_2(x)$. Finally, we assume a generalized version of shift-invariance for the inner loss function $L$:

$$\mathbb{E}_{i \sim x_2}[L(y(i), i)] = \mathbb{E}_{i \sim x_1}[L(y(i) + f^*(x_2)(i) - f^*(x_1)(i), i)] \tag{27}$$

This then implies shift-invariance for the meta-loss $\mathcal{L}$:

$$\mathcal{L}(y, x_2) = \mathcal{L}(y + f^*(x_2) - f^*(x_1), x_1) \tag{28}$$

The shift invariance assumption implies addition is defined over the action space. We believe that such additivity is reasonable for most learning problems. For instance, in classification settings, action spaces typically correspond to a vector of logits over all classes; these action spaces are Euclidean and actions can be added. Similarly, in reinforcement learning, both continuous and discrete action spaces are often additive, with the logits of a probability distribution over possible actions outputted in the discrete case. Moreover, shift-invariance is a property of the commonly used squared error loss, and indeed any loss function that has the form $\mathcal{L}(y, x) = G(y - f^*(x))$ for some function $G$.

At the same time, we note that many loss functions may not satisfy shift invariance. For example the error rate (computed as the fraction of incorrectly classified points in supervised classification), is neither differentiable nor shift-invariant. Thus, during training, differentiable proxy loss functions such as squared error or cross-entropy loss are often used, to similar effect. In other words, shift-invariant proxies can exist for non-shift-invariant losses. In reinforcement learning, loss functions may correspond to a Q function computing the cumulative negative reward over an episode conditioned on taking a particular action. In this case, loss functions often cannot be written in closed form, making it difficult to show shift invariance for these functions even if they are actually shift invariant.

For these reasons, we believe shift-invariance is a useful approximation that allows us to theoretically prove properties of task difficulty. We further note that the result of Theorem 1, which we used shift-invariance to prove, may hold even for loss functions without without shift invariance:

$$\mathcal{L}(f(x_1; \theta), x_1) - \mathcal{L}(f(x_2; \theta), x_2) \leq L_{\mathcal{L}} L_f d(x_1, x_2) d(\theta, \theta^*) \tag{29}$$

Intuitively, it states that when parameters are near their optimal values, small changes in inputs $x$ yield only small changes in the loss, and this feels like a very simple and general requirement.

## C.2 SECOND-ORDER CONDITION ON MODEL

The second order condition in Equation 4 ($\|f(x_1; \theta_1) - f(x_1; \theta_2) - f(x_2; \theta_1) + f(x_2; \theta_2)\| \leq L_f d(x_1, x_2) d(\theta_1, \theta_2)$) assumes that changes in the model $f$ can be bounded when the inputs and parameters of $f$ change a small amount. First, observe that if $f$ is twice differentiable, then the condition always holds for some choice of $L_f$ assuming $x$ and $\theta$ are defined on a bounded domain. As a concrete example, consider the case of image classification with a Tanh activated neural network where each input pixel is bounded in range $[0, 1]$ and network parameters are bounded by a large radius (the bounded parameter assumption is reasonable since practically speaking, trained neural network parameters can be bounded by some radius depending on the amount of training).

As an example of a function that does not satisfy the condition, note that neural networks with the ReLU activation function may not satisfy Equation 8. However, it has been observed that ReLU networks behave like neural networks with smooth, polynomial activation functions (Poggio et al., 2017); thus, it may be reasonable to apply our assumption even to ReLU networks.

## C.3 SHAPE OF INTERPOLATING HYPOTHESIS SPACE

Regarding our approximation of the interpolating hypothesis space, we first note that the dimensionality of the interpolating hypothesis is *lower* than the full dimensionality of the hypothesis space. Taking an interpolating hypothesis and slightly perturbing it in an *arbitrary* direction will likely break interpolation. By contrast, we expect that there exists some subspace in which perturbations along the subspace maintain interpolation (i.e. there is a continuous manifold of interpolating hypotheses). This is analogous to how in neural networks, local minima can often be in *flat* regions in which perturbing parameters along those directions does not significantly affect the loss (Keskar et al., 2017).

We approximate the interpolating hypothesis space as a disk in the overall hypothesis space (for clarity, we have revised our terminology from "ball" to "disk"); this disk has lower dimensionality than the overall hypothesis space but is embedded in a higher dimensional space. The disk assumption preserves key properties that we may reasonably expect of interpolating hypothesis spaces, namely 1) low-dimensionality and 2) boundedness, while being analytically tractable.

Finally, we emphasize that the disk approximation is made to approximate the volume of the interpolating hypothesis space. The detailed shape of the interpolating hypothesis space is not itself central to our result, but rather the scaling of its volume is more important. We believe the disk assumption captures the key scalings: the dependence of the volume on $b$ and $d$.

## C.4 EMPIRICALLY VALIDATING WASSERSTEIN DISTANCE APPROXIMATION

In this section, we empirically validate our approximation of Wasserstein distance made in Section D. Recall that We approximate the Wasserstein distance between a uniform distribution $p$ on a sphere of radius 4 and an empirical distribution $q$ of $n$ randomly sampled points on the sphere as:

$$W(p, q) \in O(rn^{-1/m}) \tag{30}$$

Note that this expression grows linearly with $r$ and decays sub-linearly with $n$, with a slower decay with larger $m$ for which more points may be required for all regions to be within a fixed radius of any point. Theoretically, we may expect this approximation to hold for large $n$ to achieve near uniform $q$. If this scaling holds, then we would expect $\log W(p, q)$ to decrease linearly with $\log n$ (with slope $-1/m$). Similarly, we would expect $\log W(p, q)$ to decrease linearly with $1/m$ (with slope $-\log n$).

We conduct experiments on a unit sphere ($r = 1$). We approximate a uniform distribution by sampling 10000 points on a unit sphere and calculate the exact Wasserstein distance between the empirical distribution of 10000 points and the empirical distribution of $n < 10000$ points (we vary $n$ from 10 to 500). Distances are computed for varying choices of $m$ varied from 3 to 19.

As illustrated in Figure 4, we find that the relationship between $\log W(p, q)$ and $\log n$ is linear with a steeper slope for small $m$, as expected. We also observe that the relationship between $\log W(p, q)$ and $\log m$ is nonlinear, with distances for large $m$ being larger than expected by a linear trend-line. Nevertheless, a linear trend-line still produces a strong fit to the empirical data.

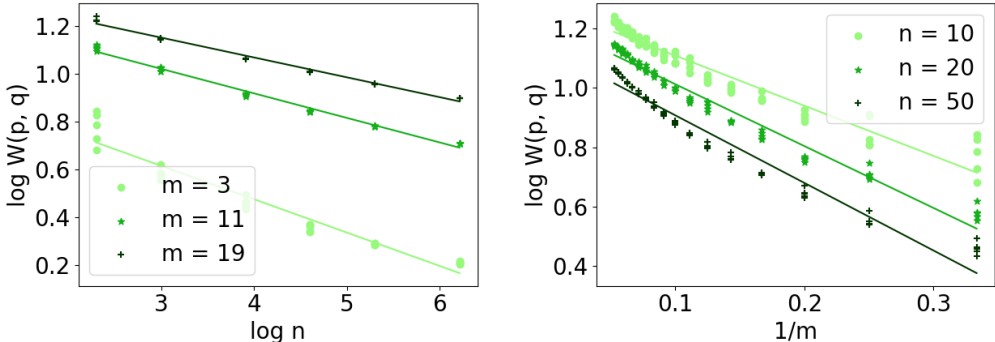

Figure 4: Wasserstein distance estimates between a uniform and empirically sampled uniform distribution on a unit sphere as a function of the dimensionality of the sphere $m$ and the number of sampled points $n$. Results are shown for five trials in each setting, and linear trend-lines are fitted to the results.

## D EMPIRICALLY APPROXIMATING TASK DIFFICULTY

In Section 4.3, we presented our inductive bias complexity approximation for supervised classification. Here, we provide more details on how to apply and compute our approximation to other settings.

First, we write a slightly modified version of our task difficulty approximation

$$\tilde{I} \approx (2dzE - nd)\left(\log b + \frac{1}{2}\log z + \log d + \log K - \frac{1}{m}\log n - \log \frac{\varepsilon}{L_{\mathcal{L}}} + \log c\right). \qquad (31)$$

where $c = \sqrt{\frac{m}{6}}\left(\frac{2\pi^{(m+1)/2}}{\Gamma\left(\frac{m+1}{2}\right)}\right)^{1/m}$, and $K$ and $E$ are functions of $m$ and the maximum frequency $M = 2\pi r/\delta$. In supervised classification settings, $z = d$.

In general settings, the parameters controlling task difficulty $(m, n, \varepsilon, L_{\mathcal{L}}, r, b, d, z)$ can be interpreted as follows: $m$ is the intrinsic dimensionality of the manifold on which instances lie. $n$ is the number of training points. $\varepsilon$ is the desired error rate on the task. $L_{\mathcal{L}}$ is the Lipschitz constant of the loss function; for tasks without an explicit, differentiable loss function (such as in reinforcement learning), it can be interpreted as the overall sensitivity of the loss with respect to the model output. $r$ and $b$ is are bound on the size of the model input and output respectively. $d$ corresponds to the dimensionality of the model output. $z$ corresponds to the number of submanifolds on which instances lie (with each combination of discrete features of instances corresponding to a submanifold). In supervised classification settings, the number of submanifolds is simply the number of classes, which is why $z$ is set equal to $d$ in these settings.

In order to compute the difficulty $\tilde{I}$ of a task, we must determine the values of these parameters. For all tasks, we set $\varepsilon/L_{\mathcal{L}}$ as the desired performance level, which is the distance in the output space corresponding to an error of $\varepsilon$. For each task, we set a fixed desired performance level corresponding to the category of the task (i.e. a fixed error rate of $1.0\%$ for image classification tasks and a fixed error rate of $0.001$ for RL tasks); see Table 3 However, because typical well-performing models in the literature trained on MNIST/ImageNet achieve error significantly better/worse performance than $1.0\%$, we adjust the desired error rate to $0.1\%$ and $10.0\%$ for MNIST and ImageNet respectively. This is done to properly quantify the difficulty of these tasks in the error rate regimes in which these tasks are typically used. However, our task difficulty numbers are relatively insensitive to the precise choice of desired performance level: using a fixed error rate of $1.0\%$ for all tasks, we find that the scale of task difficulties do not significantly change. Specifically, task difficulty falls incrementally for MNIST from $1.09 \times 10^{16}$ bits to $9.03 \times 10^{15}$ bits, and rises incrementally for ImageNet from $2.48 \times 10^{43}$ bits to $2.81 \times 10^{43}$ bits.

While intrinsic dimensionality $m$ is known for some tasks (e.g. MuJoCo (Du et al., 2018) tasks), it must be estimated for others; we use the nearest neighbors estimation approach described in Pope

Table 3: Desired error rates for different tasks used to compute a measure of inductive bias complexity. For each task, desired error rate is set as the performance of typical well-performing models proposed in the literature trained on the task. For classification tasks, error rate corresponds to test set accuracy. For Cartpole, error rate corresponds to the probability of making an incorrect action at any given time step. For MuJoCo continuous control tasks with continuous action spaces, error rate corresponds to desired distance in from the optimal action.

| Quantity | MNIST | SVHN | CIFAR-10 | ImageNet | Omniglot | Cartpole | Reacher | Hopper | Half-cheetah |
|---|---|---|---|---|---|---|---|---|---|
| Error rate | 0.1 % | 1.0 % | 1.0 % | 10.0 % | 1.0 % | 0.001 | 0.001 | 0.001 | 0.001 |

et al. (2021). The spatial resolution $\delta$ is set based on the nature of each task. For classification tasks, it is natural to set the spatial resolution as the margin between classes. For RL tasks, it is set as a scale on which trajectories with perturbations of size $\delta$ are not likely to be meaningfully different. For all tasks, we approximate $r$ to be the maximum norm of the training data points.

### D.1 EMPIRICALLY COMPUTING CLASSIFICATION TASK DIFFICULTY

For an image classification task, the model output is a probability distribution on the classes so $b$ to be 1. We set $\delta$ to be the minimum distance between points of different classes, as this is the minimum distance between inputs that $f$ must be able to distinguish. For large datasets where computing $\delta$ exactly is impractical, we estimate it using statistical methods. This procedure is explained in Appendix E.

We estimate $m$ using the technique described by Pope et al. (2021), which gives an MLE estimate based on distances to nearest neighbors. We use $k = 5$ nearest neighbors for our estimation. For large datasets where computing nearest neighbors for all points is impractical, we use the anchor approximation from Pope et al. (2021), selecting a random sample of points to use as anchors and finding their nearest neighbors in the entire dataset. To estimate $m$ for ImageNet, we randomly select 2000 points to be our anchors. See Table 4 for our estimates of $m$ and $\delta$ for image classification benchmarks.

Table 4: Estimated $m$ and $\delta$ for various image classification benchmarks. These estimates are used to estimate the amount of inductive bias required to solve each task.

| Quantity | MNIST | SVHN | CIFAR-10 | ImageNet |
|---|---|---|---|---|
| $m$ | 14 | 19 | 27 | 48 |
| $\delta$ | 2.4 | 1.6 | 2.8 | 65 |

### D.2 EMPIRICALLY COMPUTING META-LEARNING TASK DIFFICULTY

The Omniglot one-shot classification task is to identify the class of an image given an example image of each of 20 letters. Thus, the meta-learning task is to learn a function $f$ that maps a set of 20 images to a function $g$ that can map a single image to a probability distribution over 20 classes. In the following discussion, for quantities such as $z$, $d$, and $E$, a subscript $f$ will be used to denote that we are considering these values for the function $f$, and a subscript $g$ will be used to denote that we are considering these values for the function $g$.

In our framework, the input to $f$ has no discrete features, so $z_f = 1$. Since $g$ is the output of $f$, $d_f$ equals the dimensionality of the parameterization of $g$, which is $2z_g d_g E_g$. The function $g$ classifies an image among 20 classes, so $z_g = 20$ and $d_g = 19$. We can compute $E_g$ as a function of $M_g = 2\pi r_g/\delta_g$ and $m_g$. As with image classification tasks, we set $r_g$ to be the maximum norm of an image in the dataset and $\delta_g$ to be the minimum distance between images of different classes. We set $m_g$ to be the dimensionality $m_0$ of an image drawn from a single alphabet. This is done by estimating the dimensionality of each alphabet in the dataset and averaging the results.

Since the input to $f$ is 20 images, $r_f = r_g\sqrt{20}$. To compute $m_f$, let $m_1 > m_0$ be the dimensionality of an image drawn from the entire dataset. Then the dimensionality of the alphabet underlying

the input to $f$ is $m_1 - m_0$, and the dimensionality of the images themselves is $20m_0$. Therefore, $m_f = m_1 + 19m_0$. We want $f$ to be sensitive to a change in only one of the 20 images, so we set $\delta_f = \delta_g$.

The training dataset consists of several alphabets. We write $\ell_a$ for the number of letters in alphabet $a$. We set $n$ to be the number of sets of 20 images that can be drawn from the training dataset that represent 20 different letters in the same alphabet. Since the training dataset contains 20 images representing each letter,

$$n = \sum_a \binom{\max\{\ell_a, 20\}}{20} \cdot 20^{20}, \tag{32}$$

where the sum is taken over all training alphabets $a$.

Finally, $b_f$ is the maximum norm of $f$'s output, which contains the parameterization of $g$. We found that this can be bounded by $b_g \sqrt{z_g d_g}$. We have already computed $z_g$ and $d_g$, and since $g$ outputs a probability distribution, $b_g = 1$. We then use the values $z_f$, $d_f$, $m_f$, $\delta_f$, $n$, $b_f$, and the state-of-the-art error rate $\varepsilon/L_{\mathcal{L}}$ to compute the task difficulty.

### D.3 EMPIRICALLY COMPUTING REINFORCEMENT LEARNING TASK DIFFICULTY

For evaluating task difficulty for reinforcement learning tasks, we use some different approximations than for the supervised classification that are more suited to our particular setting; in particular, we use a different assumption for the manifold on which instances lie and a different approximation for Wasserstein distance $W(p, q)$.

We first write our general version of the task difficulty expression without applying the Wasserstein distance approximation in Section 4.3:

$$\tilde{I} \approx (2dzE - nd) \left( \log b + \frac{1}{2} \log z + \log d + \log K - \log r + \log W(p, q) - \log \frac{\varepsilon}{L_{\mathcal{L}}} \right). \tag{33}$$

In the reinforcement learning tasks we evaluate, $z = 1$, so we can write:

$$\tilde{I} \approx (2dE - nd) \left( \log b + \log d + \log K - \log r + \log W(p, q) - \log \frac{\varepsilon}{L_{\mathcal{L}}} \right). \tag{34}$$

The input to $f$ is an $m$-dimensional observation, which can be considered as a point in the hypercube $[-\pi, \pi]^m$ after scaling. Parameterizing $f$ as a sum of eigenfunctions of the Laplace-Beltrami operator,

$$f(x; \theta) = \sum_{p \in \mathbb{Z}^m} \theta_p e^{ip^\top x} \tag{35}$$

Treves (2016). The wavelength of the component corresponding to $p$ is $2\pi/\|p\|$, so we restrict our parameterization to components satisfying $\|p\| \leq 2\pi/\delta$. Thus, we have $L_f = 2\pi/\delta$. The number of eigenfunctions $E$ is the number of vectors $p \in \mathbb{Z}^m$ satisfying $\|p\| \leq 2\pi/\delta$, which we approximate as the volume of an $m$-dimensional ball with radius $2\pi/\delta$, giving

$$E \approx \left( \frac{2\pi}{\delta} \right)^m V_m, \tag{36}$$

where $V_m = \frac{\pi^{m/2}}{\Gamma(\frac{m}{2}+1)}$ is the volume of an $m$-dimensional ball with radius 1. We now find the Wasserstein distance $W(p, q)$. In the optimal transport, each training data point is associated with an equally-sized region of the hypercube. We approximate these regions as hypercubes with side length $s = 2\pi n^{-1/m}$. The expected distance between two randomly chosen points in one of these hypercubes is at most $s\sqrt{m/6}$ Anderssen et al. (1976), so

$$W(p, q) \approx \sqrt{\frac{m}{6}} \cdot 2\pi n^{-1/m}. \tag{37}$$

Finally, we note that $f$'s output is a force in $\{-1, +1\}$ in the discrete case and an element of $[-1, 1]^d$ in the continuous case. Therefore, we can set $b = \sqrt{d}$ in all cases. Using these values, we find

$$\tilde{I} \approx d \left( 2 \left( \frac{2\pi}{\delta} \right)^m V_m - n \right) \left( \log \frac{4\pi^2}{\sqrt{6}} + \log d + \frac{1}{2} \log m - \log \delta - \frac{1}{m} \log n - \log \frac{\varepsilon}{L_{\mathcal{L}}} \right). \tag{38}$$

Recall that $d$ is the dimensionality of the output space, and $m$ is the dimensionality of the observation space. In the noisy Cartpole task, $T > 1$ observations may be needed to determine the optimal action. Each observation is a two-dimensional state (containing the pole's angular position and velocity), so we set $m = 2T$. For fully observed tasks, note that $T = 1$.

It remains to choose values for the parameters $n$, $\delta$, and $\varepsilon/L_{\mathcal{L}}$. We choose the relatively small value of $0.001$ for $\delta$ and $\varepsilon/L_{\mathcal{L}}$ for all tasks. We choose a small value for $\delta$ because small changes in initial conditions can lead to large changes over time, and we choose a small value for $\varepsilon/L_{\mathcal{L}}$ because agents should be able to perform near-optimal actions to achieve the task's goal. In the noisy Cartpole task, we set $n = 10000$, corresponding to observing 100 episodes with 100 timesteps each. In the MuJoCo environments, we set $n = 1000000$, corresponding to observing 1000 episodes with 1000 timesteps each. We emphasize that the most important factor contributing to task difficulty is $m$, which is determined by the task specification.

## E    ESTIMATING $\delta$ FOR CLASSIFICATION TASKS

If we cannot compute $\delta$ exactly, we estimate it using extreme value theory Dekkers et al. (1989).

We wish to find the maximum value of a distribution, in this case the distribution of the reciprocal of the distance between a random pair of points from different classes. The extreme value distribution is characterized by the extreme value index $\gamma$. Let $X_1, \ldots, X_n$ be values drawn from the distribution, and let $X_{1,n} \leq \cdots \leq X_{n,n}$ be these values in sorted order. For a fixed $k < n$, define

$$M_n^{(r)} = \frac{1}{k} \sum_{i=0}^{k-1} (\log X_{n-i,n} - \log X_{n-k,n})^r. \tag{39}$$

Then we can estimate $\gamma$ as

$$\hat{\gamma}_n = M_n^{(1)} + 1 - \frac{1}{2} \left( 1 - \frac{(M_n^{(1)})^2}{M_n^{(2)}} \right)^{-1}. \tag{40}$$

Empirically, we typically find $\hat{\gamma}_n > 0$. In this case, we estimate $\delta$ by estimating the quantile corresponding to the top $1/P$ of the distribution, where $P$ is the number of pairs of training data points from different classes. By assuming that training data points are evenly distributed among the classes, we approximate $P$ as $n^2(1 - 1/C)$, where $C$ is the number of classes. If $a_n = kP/n$, this gives an estimate of

$$\frac{a_n^{\hat{\gamma}_n} - 1}{\hat{\gamma}_n} \cdot X_{n-k,n} M_n^{(1)} + X_{n-k,n} \tag{41}$$

for $1/\delta$.

The theoretical results Dekkers et al. (1989) require that $k$ and $n/k$ go to infinity as $n \to \infty$. For ImageNet, we choose $n = 40000$ and $k = 200$. To reduce the noise in our estimation, we compute 10 estimates for $1/\delta$ and average the results to obtain our final estimate for $1/\delta$.

## F    ADDITIONAL TASK VARIATIONS

In this section, in order to provide more intuition for our empirical results on inductive bias complexity, we compute the inductive bias complexity of additional variations of tasks.

### F.1    TASK COMBINATIONS

**Setup**    We first consider inductive bias complexities of task combinations. We assume we are given two tasks, corresponding to the mapping between instances $x_1$ to $y_1 = \bar{f}_1^*(x_1)$ and $x_2$ to $y_2 = \bar{f}_2^*(x_2)$ respectively. Furthermore, we assume that training and test distributions $q_1, p_1$ and $p_2, q_2$ are provided. We then construct a combination of the two tasks as the mapping from $(x_1, x_2)$ to $f^*(x_1, x_2) = (\bar{f}_1^*(x_1), \bar{f}_2^*(x_2))$. The test distribution of instances for the new task is constructed as:

$$p(x_1, x_2) = p_1(x_1) p_2(x_2) \tag{42}$$

And the training distribution constructed as:

$$q(x_1, x_2) = q_1(x_1)q_2(x_2) \tag{43}$$

Note that if $q_1$ and $q_2$ each correspond to training sets of size $n_1$ and $n_2$, $q$ corresponds to a training set of size $n_1 n_2$. We assume that the two tasks have loss functions $\mathcal{L}_1$ and $\mathcal{L}_2$ respectively. For the purposes of our analysis in this section, we assume that the loss functions satisfy $\mathcal{L}_i(y, x) \geq 0$, and equality holds if and only if $y = f_i^*(x)$ for $i = 1, 2$. We construct the loss function $\mathcal{L}$ for the combined task as:

$$\mathcal{L}((y_1, y_2), (x_1, x_2)) = \alpha \mathcal{L}_1(y_1, x_1) + (1 - \alpha)\mathcal{L}_2(y_2, x_2) \tag{44}$$

for a parameter $\alpha$ in $(0, 1)$. We denote a hypothesis as $\theta$ parameterizing a function that inputs $(x_1, x_2)$ and outputs the predictions for each task: $f(x_1, x_2; \theta) = (f_1(x_1, x_2; \theta), f_2(x_1, x_2; \theta))$. Then, the generalization error of a particular hypothesis $\theta$ under distribution $p$ (and analogous for $q$) is:

$$\begin{aligned} e(\theta, p) = \hat{e}(f(\cdot; \theta), p) &= \mathbb{E}_p[\mathcal{L}(f(x_1, x_2; \theta), (x_1, x_2))] \\ &= \alpha \mathbb{E}_p[\mathcal{L}_1(f_1(x_1, x_2), x_1)] + (1 - \alpha)\mathbb{E}_p[\mathcal{L}_2(f_2(x_1, x_2), x_2)] \end{aligned} \tag{45}$$

For notational convenience, we will also define $e_i(\theta, p)$ as (and analogous for $q$):

$$e_i(\theta, p) = \mathbb{E}_p[\mathcal{L}_i(f_i(x_1, x_2; \theta), x_i)] \tag{46}$$

for $i = 1, 2$. Thus, we may express $e(\theta, p)$ as:

$$e(\theta, p) = \alpha e_1(\theta, p) + (1 - \alpha)e_2(\theta, p) \tag{47}$$

Note that $e_1(\theta, p)$ and $e_2(\theta, p)$ *do not* correspond to errors of hypothesis of $\theta$ on tasks 1 and 2 directly because $\theta$ parameterizes a function which takes both $x_1$ and $x_2$ as input. Instead, $e_1(\theta, p)$ (and analogously for $e_2(\theta, p)$) corresponds to the error of a variant of task 1 where instances $x_1$ are augmented with a *distractor* input $x_2$ which is irrelevant to the task (producing instances $(x_1, x_2)$), but the task output $y_1$ is constructed the same way as $y_1 = \bar{f}_1^*(x_1)$. For clarify, we define this mapping as $y_1 = f_1^*(x_1, x_2)$ (and analogously for the variant of task 2). Importantly, note that this variant of task 1 has an input with a larger intrinsic dimensionality.

Now, we consider the inductive bias complexity of the combined task corresponding to $f^*$. Recall that we define inductive bias complexity as:

$$\tilde{I} = -\log \mathbb{P}(e(\theta, p) \leq \varepsilon \mid e(\theta, q) = 0) \tag{48}$$

We relate this inductive bias complexity to the inductive bias complexities of tasks corresponding to $f_1^*$ and $f_2^*$:

$$\tilde{I}_i = -\log \mathbb{P}(e_i(\theta, p) \leq \varepsilon \mid e_i(\theta, q) = 0) \tag{49}$$

where $\tilde{I}_i$ corresponds to the inductive bias complexity for the task corresponding to $f_i^*$. Under certain conditional independence assumptions, we are able to relate $\tilde{I}$ and the $\tilde{I}_i$. Specifically, we assume that if $\theta$ interpolates task 1, then additionally interpolating task 2 does not change the probability that $\theta$ will generalize on task 1 (and vice versa). We also assume that if $\theta$ interpolates both task 1 and task 2, then generalizing on task 1 does not affect the probability that $\theta$ generalizes on task 2 (and vice versa). This is reasonable if we consider task 1 and task 2 to be constructed independently in the sense that inputs $x_2$ do not provide information on $f_1^*(x_1, x_2)$ and vice versa; being able to interpolate or generalize on one task does not affect the ease of generalization on the other. Thus, we are able to make the following statement:

**Theorem 2.** *Assume the following conditional independencies:*

$$e_1(\theta, p) \leq \varepsilon \perp\!\!\!\perp e_2(\theta, q) = 0 | e_1(\theta, q) = 0 \tag{50}$$

$$e_2(\theta, p) \leq \varepsilon \perp\!\!\!\perp e_1(\theta, q) = 0 | e_2(\theta, q) = 0 \tag{51}$$

$$e_1(\theta, p) \leq \varepsilon \perp\!\!\!\perp e_2(\theta, p) \leq \varepsilon | e_1(\theta, q) = 0, e_2(\theta, q) = 0 \tag{52}$$

*Then,*

$$-\log(e^{-\tilde{I}_1} + e^{-\tilde{I}_2}) \leq \tilde{I} \leq \tilde{I}_1 + \tilde{I}_2 \tag{53}$$

*Proof.* First, that by conditional independence:

$$\mathbb{P}(e_1(\theta, p) \leq \varepsilon \mid e(\theta, q) = 0)\mathbb{P}(e_2(\theta, p) \leq \varepsilon \mid e(\theta, q) = 0)$$
$$= \mathbb{P}(e_1(\theta, p) \leq \varepsilon \wedge e_2(\theta, p) \leq \varepsilon \mid e(\theta, q) = 0) \quad (54)$$

Note that $e(\theta, p) \leq \varepsilon \impliedby e_1(\theta, p) \leq \varepsilon \wedge e_2(\theta, p) \leq \varepsilon$. This implies:

$$\mathbb{P}(e(\theta, p) \leq \varepsilon \mid e(\theta, q) = 0) \geq \mathbb{P}(e_1(\theta, p) \leq \varepsilon \wedge e_2(\theta, p) \leq \varepsilon \mid e(\theta, q) = 0)$$
$$= \mathbb{P}(e_1(\theta, p) \leq \varepsilon \mid e(\theta, q) = 0)\mathbb{P}(e_2(\theta, p) \leq \varepsilon \mid e(\theta, q) = 0) \quad (55)$$

Also, observe that $e(\theta, p) \leq \varepsilon \implies e_1(\theta, p) \leq \varepsilon \vee e_2(\theta, p) \leq \varepsilon$. This implies:

$$\mathbb{P}(e_1(\theta, p) \leq \varepsilon \mid e(\theta, q) = 0) + \mathbb{P}(e_2(\theta, p) \leq \varepsilon \mid e(\theta, q) = 0) \geq \mathbb{P}(e(\theta, p) \leq \varepsilon \mid e(\theta, q) = 0) \quad (56)$$

Next, using our first two conditional independence assumptions:

$$\mathbb{P}(e_1(\theta, p) \leq \varepsilon \mid e_1(\theta, q) = 0)\mathbb{P}(e_2(\theta, p) \leq \varepsilon \mid e_2(\theta, q) = 0) \leq \mathbb{P}(e(\theta, p) \leq \varepsilon \mid e(\theta, q) = 0) \quad (57)$$

$$\mathbb{P}(e_1(\theta, p) \leq \varepsilon \mid e_1(\theta, q) = 0) + \mathbb{P}(e_2(\theta, p) \leq \varepsilon \mid e_2(\theta, q) = 0) \geq \mathbb{P}(e(\theta, p) \leq \varepsilon \mid e(\theta, q) = 0) \quad (58)$$

Finally, taking the negative log of both sides:

$$-\log(e^{-\tilde{I}_1} + e^{-\tilde{I}_2}) \leq \tilde{I} \leq \tilde{I}_1 + \tilde{I}_2 \quad (59)$$

as desired. □

Note that $-\log(e^{-\tilde{I}_1} + e^{-\tilde{I}_2}) \approx \min\{\tilde{I}_1, \tilde{I}_2\}$. Thus, the statement intuitively says that the combined task requires inductive bias up to the total inductive bias of the two tasks treated individually, and requires *at least* the inductive bias of the easier task. We emphasize again that $\tilde{I}_i$ does *not* correspond to the inductive bias required to solve task $i$, but rather the inductive bias required to solve a variant of task $i$ with a distractor added to instances provided from the other task.

**Experiments**  Next, we empirically compute inductive bias complexities for combinations of image classification tasks. To do this, we compute the task difficulty of a version of each task with a task-irrelevant distractor from the *other* task appended to each instance ($\tilde{I}_1$ or $\tilde{I}_2$ as described above). We then report the upper bound $\tilde{I}_1 + \tilde{I}_2$ and lower bound $-\log(e^{-\tilde{I}_1} + e^{-\tilde{I}_2})$ of $\tilde{I}$.

In order to compute task difficulties for tasks with distractor-appended instances, we first revisit our generalized approximation of inductive bias complexity from Equation 31:

$$\tilde{I} \approx (2dzE - nd)\left(\log b + \frac{1}{2}\log z + \log d + \log K - \frac{1}{m}\log n - \log\frac{\varepsilon}{L_{\mathcal{L}}} + \log c\right). \quad (60)$$

where $c = \sqrt{\frac{m}{6}}\left(\frac{2\pi^{(m+1)/2}}{\Gamma\left(\frac{m+1}{2}\right)}\right)^{1/m}$, which is roughly constant for large intrinsic dimensionality $m$. Recall that $d$ is the output dimensionality, $z$ is the number of manifolds that the input data lie on, $n$ is the number of training data points, $\varepsilon$ is the target error, $L_{\mathcal{L}}$ is the Lipschitz constant of the loss function, and $K$ and $E$ are functions of $m$ and the maximum frequency $M = 2\pi r/\delta$. Typically for classification problems, $z = d$ is set as the number of classes.

When we append task-irrelevant distractors to the instances of a task, five key parameters change: the number of training points $n$, the intrinsic dimensionality $m$, the number of manifolds $z$ and the bound on model output $b$ and bound on model input $r$. Importantly, all other parameters stay fixed (namely, $\delta$, $d$, $\varepsilon$ and $L_{\mathcal{L}}$). The number of training points scales with the number of different distractor instances; if task 1 with $n_1$ training points is augmented with instances from task 2 with $n_2$ training points, the new task has $n_1 n_2$ training points. Also, the new intrinsic dimensionality of the task is the *sum* of the intrinsic dimensionalities of the individual tasks since the combined manifold of $(x_1, x_2)$ is the *product* of the manifolds of $x_1$ and $x_2$ individually. Thus, if task 1 has intrinsic dimensionality $m_1$ and task 2 has intrinsic dimensionality $m_2$, the intrinsic dimensionality of the augmented task is $m_1 + m_2$. The number of manifolds of the input of the combined task is the number of manifolds corresponding to $(x_1, x_2)$, which is simply $z_1 z_2$ if task 1 and task 2 have $z_1$ and $z_2$ classes respectively. If task 1 and task 2 have model outputs bounded by $b_1$ and $b_2$ respectively,

then the combined model is bounded by $\sqrt{b_1^2 + b_2^2}$. Similarly, the new value of $r$ is $\sqrt{r_1^2 + r_2^2}$. We may then compute the task difficulty of distractor-appended task using the formula above.

We compute task difficulties for pairwise combinations of MNIST, SVHN and CIFAR-10 (which have individual task difficulties (in bits) of: $1 \times 10^{16}$, $1 \times 10^{31}$, $3 \times 10^{32}$). As found in Table 5, combined task difficulties are significantly greater than the difficulties of individual tasks. As a very rough rule of thumb, the combined task difficulty is approximately the product of the individual task difficulties. This makes sense since the task difficulty scales exponentially with the intrinsic dimension of the data and combining together two tasks *adds* together the intrinsic dimensionality of the two tasks (and thus multiplies their task difficulties).

Table 5: Combined task difficulty bounds (in bits) for combinations of image classification tasks. Difficulties are reported as "lower bound / upper bound." Individual task difficulties of MNIST, SVHN and CIFAR-10 are (in bits): $1 \times 10^{16}$, $1 \times 10^{31}$, $3 \times 10^{32}$ respectively.

|  | MNIST | SVHN | CIFAR-10 |
|---|---|---|---|
| MNIST | $1 \times 10^{26}$ / $3 \times 10^{26}$ | $6 \times 10^{39}$ / $5 \times 10^{45}$ | $3 \times 10^{42}$ / $8 \times 10^{44}$ |
| SVHN | - | $4 \times 10^{54}$ / $8 \times 10^{54}$ | $4 \times 10^{50}$ / $4 \times 10^{61}$ |
| CIFAR-10 | - | - | $2 \times 10^{55}$ / $3 \times 10^{55}$ |

Why does combining together two simple tasks like MNIST result in a much more difficult task? This is because separating the two parts of the combined task requires a lot of inductive bias. Without this inductive bias, solving a combination of two tasks looks like solving a single task with a higher dimensional input, which correspondingly requires much greater inductive bias. This also suggests how practical model classes like neural networks may be able to provide such vast amounts of inductive bias to a task: namely, by breaking down the input into lower-dimensional components.

### F.2 PREDICTING ZERO-OUTPUT

To gain intuition for the properties of task difficulty, we compute task difficulties for a task in which the target function always has output 0: $f^*(x)$. To compute the task difficulty, we return to the definition of inductive bias complexity from Definition 1:

$$\tilde{I} = -\log \mathbb{P}(e(\theta, p) \leq \varepsilon \mid e(\theta, q) \leq \epsilon) \tag{61}$$

Keeping $\epsilon = 0$ as we have done throughout the paper, observe that just as with the general case, the inductive bias complexity is based on the fraction of interpolating hypotheses that generalize well. In the case of zero output, generalizing well means that $f(\cdot; \theta)$ should be sufficiently close to 0 on the distribution $p$. Note that interpolating hypotheses may not necessarily have near-zero output over distribution $p$; thus, solving the a zero-output task may require significant levels of inductive bias.

The key to determining the difficulty of a zero-output task is setting the hypothesis space. Recall that in Section 4.3, we constructed the hypothesis space to be a linear combinations of a set of basis functions sensitive to a *task-relevant* resolution (or larger). In the case of zero-output, all resolutions are *task-irrelevant*: the true output $f^*(x)$ is not sensitive to any changes in the input. Thus, if we were to use the approach of Section 4.3 to construct hypotheses, we would find that there is only a single hypothesis in the hypothesis space, namely $f*$. Since $f^*$ interpolates the training data and perfectly generalizes, the probability of generalizing given interpolation is 1. Thus, the task difficulty is 0.

However, other choices of the base hypothesis space may lead to very large task difficulties. Consider a version of ImageNet in which all inputs are mapped to a 1000-dimensional zero vector. If we construct the base hypothesis space in the same way as for the original ImageNet task, assuming we wish to achieve the same generalization error target as we set for the original ImageNet task, the task difficulty approximation for the zero-output ImageNet would be the *same* as for the original ImageNet: $3.55 \times 10^{41}$ bits.

How can the inductive bias required to generalize on ImageNet stay the same even when the target function is made much simpler (assuming the hypothesis space is kept the same)? This is because

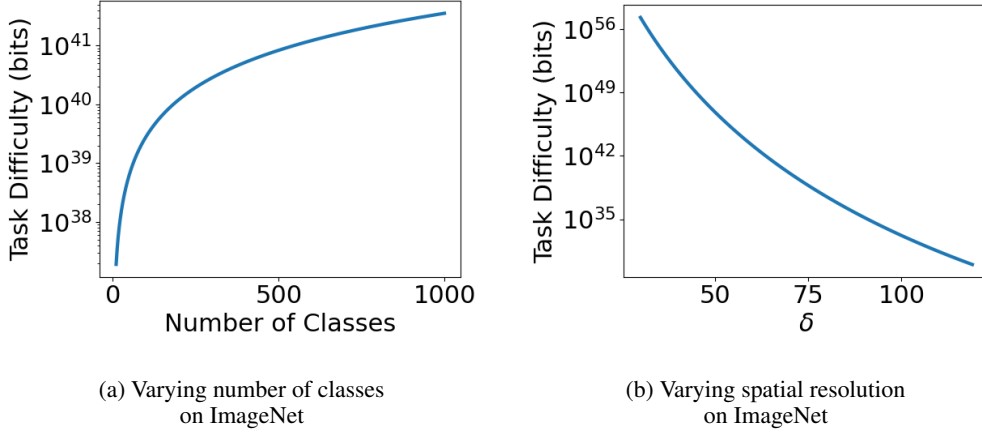

(a) Varying number of classes
on ImageNet

(b) Varying spatial resolution
on ImageNet

Figure 5: Task difficulties on parametric variations of benchmark tasks.

inductive bias complexity corresponds to the difficulty of specifying a generalizing set of hypotheses from the interpolating hypotheses. Due to the large size of the base hypothesis space, the size of the interpolating hypothesis space can be expected to be the same for both the true ImageNet target function and the zero target function: we may expect a the same fraction of hypothesis to satisfy the zero target training samples as the true ImageNet training samples. The base hypothesis space does not over-represent hypotheses near the zero target function relative to other hypotheses. Then, given similarly sized interpolating hypothesis spaces, the difficulty of specifying the true hypothesis is similar for the two tasks.

### F.3 PREDICTING RANDOM TARGETS

Next, we consider a variation of ImageNet where the ImageNet target function is replaced with a random function selected from the base hypothesis space used for the original ImageNet. In this case, the training set would appear to be a version of ImageNet with randomized outputs for each image. Importantly, note that this is different than independently selecting a random output for each point in the training set: we would training points that are sufficiently close (with distances around the spatial resolution $\delta$ or smaller) to have similar outputs. We consider this formulation of randomizing the outputs of ImageNet instead of selecting independent random targets for each ImageNet input since constructing independently chosen random targets for all possible input of ImageNet may not correspond to a well-defined target function.

Assume we wish to achieve the same generalization error rate as for the original ImageNet task. The parameters governing task difficulty would remain the same as in Equation 11: namely, the parameters specific to the input distribution $(m, n, d, r)$, model output and loss function $(b, L_{\mathcal{L}})$ and error rate $(\varepsilon)$ would remain the same. Thus, the task difficulty would be the *same* as for the original ImageNet: $3.55 \times 10^{41}$ bits.

How can solving a randomized version of ImageNet require no more inductive bias than the original task despite having a less structured target function? Intuitively, this is because the base hypothesis space considers all hypothesis equally: it does not over-represent structured hypotheses (such as the target function of ImageNet) relative to other ones. Thus, the difficulty of specifying a well generalizing region in the hypothesis space is the same for both the randomized and original versions of ImageNet.

### F.4 VARYING NUMBER OF CLASSES

We vary the number of classes in ImageNet (while keeping all other task parameters fixed including the number of training points $n$) and find in Figure 5a that task difficulty grows with the number of classes. As the number of classes increased from 10 to 1000, the task difficulty increases by roughly 5 orders of magnitude. This is primarily due to the increased size of the input space when more classes

are added. The results indicate that adding more classes to a classification task can be a moderately powerful way of increasing the difficulty of a task, although not as powerful as increasing the intrinsic dimensionality of a task.

### F.5  Varying Spatial Resolution

We vary the spatial resolution used to construct the base hypothesis space of ImageNet (while keeping all other task parameters fixed) and find in Figure 5b that task difficulty drastically shrinks with the spatial resolution. This makes sense: as the spatial resolution used to construct the hypothesis space grows, the hypothesis space shrinks, and it becomes much more difficult to specify regions in the hypothesis space. These results emphasize the critical effect of the selection of base hypothesis space on task difficulty.

## G  Additional Discussion

### G.1  How to extend the inductive bias complexity measure to the non-interpolating case?

One limitation of our work is that we only consider interpolating hypotheses (in other words, we only consider training error $\epsilon = 0$ in Definition 1). Although we will leave a formal extension of our results to non-interpolating hypotheses as a future work, we briefly outline how are results might be extended:

First, we may provide an analogous result to Theorem 1 in the non-interpolating case by arguing that non-interpolating hypotheses may generalize well as long as they are within a smaller radius of the true hypotheses (relative to the radius for interpolating hypotheses). Specifically, we may expect the more general expression for the radius to be $\frac{\varepsilon - \epsilon}{L_{\mathcal{L}} L_f W(p,q)}$.

Next, in order to quantify probabilities in the hypothesis space and find a practical estimate for inductive bias complexity, we must quantify the size of the hypothesis space that fits the training data up to error $\epsilon$. Intuitively, we may expect it to be a region around the interpolating hypothesis space with dimensionality equal to that of the base hypothesis space. We may expect the size of this region to scale polynomially with $\epsilon$. As with the interpolating case, we can then estimate the fraction of the near-interpolating hypothesis space that is close enough to the true hypothesis to find a generalized, practically-computable expression for task difficulty.

### G.2  Why does inductive bias exponentially depend on data dimension?

Intuitively, inductive bias scales exponentially with data dimension because inductive bias scales with the dimensionality of the hypothesis space, and the dimensionality of the hypothesis space scales exponentially with data dimension.

First, we consider the intuition for why inductive bias scales with the dimensionality of the hypothesis space: recall that generalizing requires specifying a specific set of well-generalizing hypotheses in the hypothesis space. The amount of information needed to specify a point in an $m$ dimensional space scales with $m$ since it is simply the number of coordinates of the point. Thus, assuming well-generalizing hypotheses are concentrated in a region around a point, the amount of information required to specify the region also scales with the dimensionality of the hypothesis space.

Next, we consider the intuition for why the dimensionality of the hypothesis space scales with data dimensionality: consider a very simple "grid-based" method of constructing hypotheses in which the input manifold is divided into equally sized hypercubes that tile the entire manifold. A hypothesis consists of a mapping between hypercubes and outputs. Note that the dimensionality of each hypothesis scales with the number of hypercubes since each hypothesis can be specified by its output at each hypercube. The number of hypercubes scales exponentially with the dimensionality of the input manifold; thus the hypothesis space dimensionality also scales exponentially with the dimensionality of the input manifold.

We view this exponential dependence as fundamental: generalizing over more dimensions of variation significantly expands the hypothesis space regardless of how we parameterize hypotheses, and thus dramatically increases inductive bias complexity.

### G.3 WHY DO TRAINING POINTS PROVIDE SO LITTLE INDUCTIVE BIAS?

Experimentally, we observe that changing the amount of training data, even by many orders of magnitude, does not significantly affect the amount of inductive bias required to generalize. We can interpret this as training data providing relatively little information on which to generalize. Why do training points provide so little information? Intuitively, it is because in the absence of strong constraints on the hypothesis space, training points only provide information about the function we aim to approximate in a *local* neighborhood around each training point. By contrast, inductive biases can provide more *global* constraints on the hypothesis space than can dramatically reduce the size of the hypothesis space and allow for generalization. Without strong inductive biases, training points need to cover large regions of the input manifold to allow for generalization. With high dimensional manifolds, this can require very large numbers of training points.

Through simple scaling arguments, we can estimate the number of training samples we need to generalize on ImageNet in the absence of strong inductive biases. We estimate the intrinsic dimensionality of ImageNet to be $48$ and the task-relevant resolution $\delta$ of ImageNet to be $65$. We can then estimate the number of training points needed to generalize as the number such that any region on the manifold is within radius $\delta$ of a training point. With $n$ training points, we can expect to cover a volume of about $n65^{48}$ on the manifold. Given a manifold of volume about $(255 \times 224 \times \sqrt{3})^{48}$ (corresponding to the $[0, 255]$ pixel range and $224 \times 224 \times 3$ extrinsic dimensionality of ImageNet images), we would then require about $(255 \times 224 \times \sqrt{3}/65)^{48} \approx 10^{153}$ points to generalize. Indeed, following the linear trend of Figure 3, we may expect to approximately halve the required inductive bias with this number of training points. However, with only $10^{17}$ points, strong inductive biases would be necessary to generalize as indicated in Figure 3.

### G.4 WHY IS THE SCALE OF INDUCTIVE BIAS COMPLEXITY SO LARGE?

Across all settings, our measure yields very large inductive information content, many orders of magnitude larger than parameter dimensionalities of practical models. These large numbers can be attributed to the vast size of the hypothesis space constructed in Section 4.3: it includes *any* bandlimited function on the data manifold below a certain frequency threshold. Typical function classes may already represent only a very small subspace of the hypothesis space: for instance, neural networks have biases toward compositionality and smoothness that may significantly reduce the hypothesis space (see Li et al. (2018); Mhaskar et al. (2017)), with standard initializations and optimizers further shrinking the subspace. Moreover, functions that can be practically implemented by reasonably sized programs on our computer hardware and software may themselves occupy a small fraction of the full hypothesis space. Future work may use a base hypothesis space that already includes some of these constraints, which could reduce the scale of our measured task difficulty.

## H  ADDITIONAL TABLES AND FIGURES

Table 6: Mapping notation in the feature-based task framework to different learning settings.

| Setting | $x$ | Features | $f^*(x)$ |
|---|---|---|---|
| **Supervised classification** | input | $d_1$ : class
$c_1$ : instance within class | class |
| **Reinforcement learning** | state or observation sequence | $c_1$ : state or observation sequence | desired action |
| **Meta-learning for few-shot classification** | samples from a set of related classes | $c_1$: a set of related classes
$c_2$: choice of samples from $c_1$ | classifier mapping inputs to their class |
| **Unsupervised autoencoding** | input | $c_1$: input | $x$ |
| **Meta-reinforcement learning** | trajectories through environments | $c_1$: environment
$c_2$: trajectory within environment | policy mapping states to desired actions |

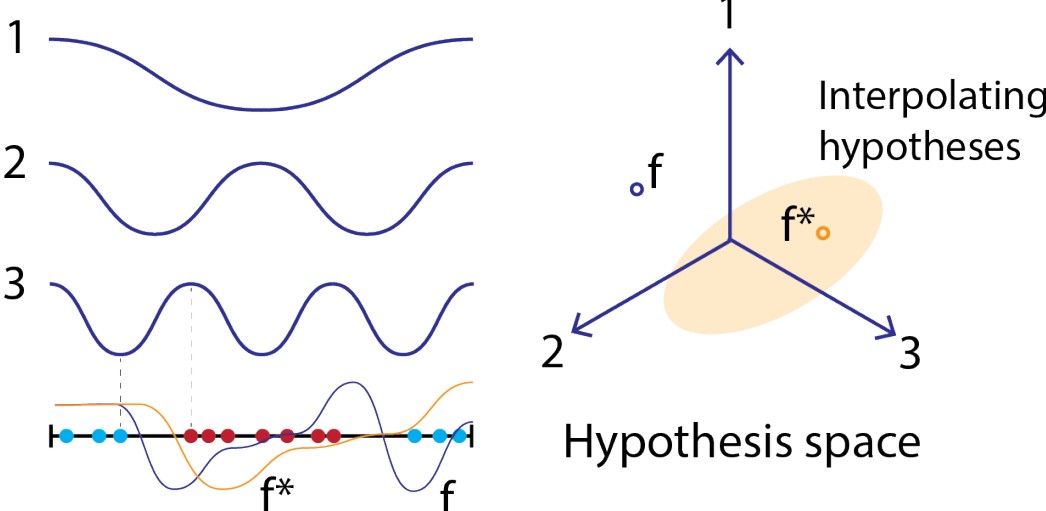

Figure 6: An example of constructing a hypothesis space for a binary-classification task with a 1 dimensional input. The black line indicates the data manifold and the blue and red dots on the line indicate training points. The hypothesis space is constructed using basis functions of three different frequencies; observe that the highest frequency is chosen to have scale corresponding to the minimum distance between classes. Two specific hypothesis are illustrated in the hypothesis space, the true hypothesis (in orange), $f^*$ and another hypothesis $f$ (in purple). Both can be expressed as linear combinations of the basis functions, and thus correspond to points in the hypothesis space as illustrated. The $f*$ hypothesis fits the training data, and thus is part of the interpolating hypothesis set indicated by the light orange oval.

Table 7: The inductive bias information contributed by different model architectures for image classification tasks. (FC-$N$ refers to a fully connected network with $N$ layers.)

**MNIST**

| Model | Information Content ($\times 10^{16}$ bits) |
|---|---|
| Linear (Lecun et al., 1998) | 0.705 |
| FC-3 (Lecun et al., 1998) | 0.817 |
| AlexNet (mrgrhn, 2021) | 0.888 |
| LeNet-5 (CNN) (Lecun et al., 1998) | 0.907 |
| DSN (Lee et al., 2015) | 0.976 |
| MCDNN (Ciregan et al., 2012) | 1.020 |
| Ensembled CNN (An et al., 2020) | 1.094 |

**SVHN**

| Model | Information Content ($\times 10^{31}$ bits) |
|---|---|
| FC-6 (Mauch & Yang, 2017) | 0.870 |
| AlexNet (Veeramacheneni et al., 2022) | 0.985 |
| Deep CNN (Goodfellow et al., 2013) | 1.049 |
| DSN (Lee et al., 2015) | 1.060 |
| DenseNet (Huang et al., 2017) | 1.075 |
| WRN-16-8 (Zagoruyko & Komodakis, 2016) | 1.077 |
| WRN-28-10 (Foret et al., 2020) | 1.114 |

**CIFAR10**

| Model | Information Content ($\times 10^{32}$ bits) |
|---|---|
| Linear (Nishimoto, 2018) | 2.250 |
| FC-4 (Lin et al., 2015) | 2.354 |
| MCDNN (Ciregan et al., 2012) | 2.704 |
| AlexNet (Krizhevsky et al., 2012) | 2.709 |
| DSN (Lee et al., 2015) | 2.786 |
| DenseNet (Huang et al., 2017) | 3.010 |
| ResNet-50 (Wightman et al., 2021) | 3.195 |
| BiT-L (Kolesnikov et al., 2020) | 3.453 |
| ViT-H/14 (Dosovitskiy et al., 2021) | 3.513 |

**ImageNet**

| Model | Information Content ($\times 10^{41}$ bits) |
|---|---|
| Linear (Karpathy, 2015) | 2.063 |
| SIFT + FVs (Sánchez & Perronnin, 2011) | 2.261 |
| AlexNet (Krizhevsky et al., 2012) | 2.290 |
| DenseNet-121 (Huang et al., 2017) | 2.348 |
| ResNet-50 (He et al., 2016) | 2.362 |
| DenseNet-201 (Huang et al., 2017) | 2.363 |
| WRN-50-2-bottleneck (Zagoruyko & Komodakis, 2016) | 2.368 |
| BiT-L (Kolesnikov et al., 2020) | 2.449 |
| ViT-H/14 (Dosovitskiy et al., 2021) | 2.461 |

