# OpenReview forum: "Model-agnostic Measure of Generalization Difficulty"
_ICLR.cc/2023/Conference — Submitted to ICLR 2023_

### Official Review · Reviewer_Du5K · 2022-10-19

**Confidence:** 4
**Correctness:** 3
**Technical Novelty And Significance:** 2
**Empirical Novelty And Significance:** Not applicable
**Recommendation:** 3

**Clarity, Quality, Novelty And Reproducibility:**

The paper is mostly clear. The proof of Theorem 1 is easy to follow. However, the paper does not use notations consistently, which makes it a bit difficult in a few places. For example, the authors use small letters for functions $f$ but capital letters for the loss function $L$, while using small letters for dimensionality of data and size but not dimensionality of output, etc.

In a few places, the authors introduce complicated definition to describe simple concepts that are already well-established in the statistical learning theory literature. For example, the discussion in Section 3.1 corresponds basically to the "general setting of learning", which was pioneered by Vapnik [4].

In addition, the authors refer to "inductive bias complexity" repeatedly in the paper before it was formally defined in Page 5. It would be helpful to define it first, at least informally, so that the discussions in Sections 1, 2, and 3 can be easier to follow.

[4] V. Vapnik, “An overview of statistical learning theory,” IEEE TNN, 1999.

**Strength And Weaknesses:**

*Strengths*:
- The authors pursue a novel direction, which is to provide a model-agnostic measure of task difficulty. However, the final outcome (Equation 11) is quite similar to several recent works that study scaling laws, which use much simpler arguments.

*Weaknesses*:
- Under the hood, "inductive bias complexity" corresponds to the fraction of interpolating hypotheses that also generalize well; i.e. the probability that an interpolating hypothesis would also generalize. When this is taken as a definition, we obviously have that interpolating functions would generalize well on simple tasks and perform poorly on more difficult tasks; but this is really somewhat tautological. I don't think the definition offers any insight that can be useful in practice.
- The authors argue that the definition is model-agnostic. However, the definition does assume that one can (somehow) parameterize the space of all possible hypotheses.
- If we consider simpler arguments used in the scaling laws literature, we deduce that difficulty of a task is roughly $O(m^{-1/d})$, where $m$ is the size of the training set and $d$ is the intrinsic dimension of the data manifold. This follows from a partitioning argument; see for example [1, 2, 3]. It's not clear if the proposed "inductive bias complexity" in Theorem 1 would offer any additional insight beyond such simpler measures of complexity. In fact, the series of approximations that the authors propose are similar in spirit to previous arguments and also lead to similar conclusions (see for example Equation 11).
- The definition that the authors use involves a loss $\epsilon$. In Appendix C, the authors use different values of  $\epsilon$ for different tasks (e.g. they use a very small value in MNIST and a much larger value in ImageNet) even though they also show that "inductive bias" is sensitive to the choice of this parameter (Figure 3b). I'm not sure if this makes the comparison between datasets meaningful.

[1] Yasaman Bahri, Ethan Dyer, Jared Kaplan, Jaehoon Lee, and Utkarsh Sharma. Explaining neural scaling laws. arXiv preprint arXiv:2102.06701, 2021.

[2] Marcus Hutter. Learning curve theory. arXiv preprint arXiv:2102.04074, 2021.

[3] Utkarsh Sharma and Jared Kaplan. Scaling laws from the data manifold dimension. JMLR, 23(9):1–34, 2022.


**Summary Of The Paper:**

The authors introduces a notion of task difficulty, which they coin "inductive bias complexity". Intuitively, it is the fraction of interpolating hypotheses (in the space of all possible functions) that also generalize well. After introducing the formal definition, the authors use a series of approximations to come up with a computationally tractable approximation. Then, they apply it on various tasks and argue that this measure of complexity matches with expectations; e.g. MNIST < CIFAR10 < ImageNet.

**Summary Of The Review:**

I think the direction is novel but the approach the authors use is very similar to recent works that study scaling law estimators. In that line of work, the goal is to understand power laws in overparaemterized models. In this work, the goal is to quantify the difficulty of a task but the approach is questionable for several reasons as I mention above. The paper would benefit from providing a stronger motivation that highlights clearly how this measure of complexity can be useful to the community (either by providing new insights or practice applications).

---

> ### Author Response · Authors · 2022-11-18
> **Authors' Response to Reviewer (1/?)**
>
> **Concerns on similarity of our result with existing scaling laws**
>
> We first thank the reviewer for pointing out the similarity between the result of our work and the scaling laws found for sample complexity in previous literature. Indeed, simpler arguments in prior literature show that the sample complexity of a task scales as $O(1/\varepsilon^m)$ where $\varepsilon$ is the desired error rate and $m$ is the intrinsic dimensionality of the data; note the exponential dependence on intrinsic data dimensionality.
> In contrast to this prior literature which provides scaling for sample complexity (or alternatively shows how generalization error scales as a function of the number of samples), we provide a scaling law for *inductive bias complexity.* It is not surprising that sample complexity scales similarly to inductive bias complexity; indeed, both quantify the amount of information required to generalize with the former quantifying the information provided by the training set and the latter quantifying the information provided by the model designer. However, to our knowledge, we are the first to show any scaling trends for inductive bias complexity.
>
> Please see our general response for additional discussion on the practical value of having a measure of inductive bias complexity.
>
> Thank you for the additional references; we have added a discussion of them in our revised Section 2.
>
> **Concerns on definition of task difficulty**
>
> Indeed, as the reviewer notes, we have defined inductive bias complexity based on the fraction of interpolating hypotheses that generalize well. We then use this inductive bias complexity measure as a way of quantifying the difficulty of generalizing on a task. Our experiments demonstrate that tasks which we may intuitively view as more difficult indeed can be quantified as requiring greater inductive bias.
>
> However, our measure does not merely recapitulate human intuition for which tasks are more difficult than others. On the contrary, we find that defining task difficulty as inductive bias complexity yields some unintuitive insights:  e.g. simple tasks with many degrees of variation (e.g. ImageNet) are more difficult than complex tasks with few degrees of variation (e.g. Omniglot). Please see our general response for further details on the practical applicability of our measure of inductive bias complexity.
>
> We more formally define inductive bias in Section 4.1 our revision; see our general response for more details.
>
> **Concerns on model-agnosticism of framework**
>
> As the reviewer notes, we must parameterize the space of possible hypotheses in order to quantify inductive bias complexity. However, as we emphasize in Section 4, this parameterization of hypothesis is *not* related to the model classes used to actually train on a task. Instead, the parameterization of the base hypothesis space is chosen to encompass all reasonable model classes that might be applied to a task.
>
> Importantly, we *do* assume all relevant hypotheses can be parameterized in our base hypothesis space. We believe this assumption is reasonable due to the vast size of the base hypothesis space as illustrated by our large numbers for inductive bias complexity; the hypothesis space includes *any* bandlimited function up to a certain frequency. Please see Section 6 for further discussion.
>
> **Concerns on sensitivity to error rate**
>
> As we describe in Appendix D, we select different target error rates for different categories of tasks because 1) performance metrics for different tasks (such as RL and classification tasks) are often not comparable and 2) typical performance levels reported for different tasks may be quite different (for example, for MNIST vs. ImageNet).
>
> Nevertheless, note that desired error rate does *not* significantly affect the scale of task difficulty as found in Figure 3. We also recompute task difficulties for *all* image classification tasks using a fixed error rate of $1.0 \%$ and find that the scale of task difficulties do not significantly change: $1.09 \times 10^{16}$ bits to $9.03 \times 10^{15}$ bits for MNIST, and $2.48 \times 10^{43}$ bits to $2.81 \times 10^{43}$ bits for ImageNet.
>
> We have included these results in our revised Appendix D.

---

> > ### Author Response · Authors · 2022-11-18
> > **Authors' Response to Reviewer (2/?)**
> >
> > **Concerns on novelty of feature-based task framework**
> >
> > We agree with the reviewer that prior literature has already established that many different learning problems across domains have the same generalization structure; indeed, we note in Section 3.2 that previous literature already establishes that generalization in RL and supervised learning can be viewed in a similar way. Our goal in Section 3 is not to produce a novel framework to demonstrate that many different learning problems (supervised classification, RL, meta-learning etc.) have the same generalization structure. Rather, it is to provide a unified set of notation on top of which we may define the concept of inductive bias complexity (in Section 4.1) and propose a measure to compute it (in Section 4.2 and 4.3).
> >
> > In our revised Section 3, we have clarified the motivation for our feature based task framework and emphasized the contributions of prior work.

---

> > ### Comment · Reviewer_Du5K · 2022-11-23
> > **Response**
> >
> > Thank you for the response. Using a fixed error rate of 1% in Appendix D answers one question. But, the main issues remain there.
> >
> > As I mentioned above, the relation between your definition of inductive bias and the ease of a task is tautological; you define inductive bias by how easy a task is so it's natural that easy tasks have a small inductive bias. In your response, you mention that you discuss scaling for inductive bias but (again) this is somewhat equivalent to the scaling of loss studied in recent works because of the nature of the definition.
> >
> > In addition, the fact that you require that all hypotheses be parameterized means *by definition* that your measure is not model agnostic. It is calculated according to a chosen method of parameterizing all hypotheses, which implies that you do assume a fixed hypothesis space.
> >
> > To emphasize on the positive side, I do think that the goal pursued in this work (i.e. quantifying inductive bias) is useful, but I doubt that the approach proposed here would be useful, though.

---

> > > ### Author Response · Authors · 2022-11-24
> > > **Thank you for your response! (1/2)**
> > >
> > > We appreciate the reviewer's engagement with our response. We address the remaining concerns as follows:
> > >
> > > **Concerns on definition of inductive bias**
> > >
> > > We would like to clarify the different notions of difficulty or complexity being considered here. In our paper, we have defined "inductive bias complexity" based on the "fraction of interpolating hypotheses that generalize well." This is distinct from three other notions: 1) "how easy a task is" to a human trying to solve a task, 2) "how easy a task is" to a researcher trying to develop a model to generalize on a task, 3) "how easy a task is" to a model class as measured by its generalization error given a specific amount of training data. Notions 1 and 2 are not directly quantifiable, and we do not aim to quantify them in our paper. Notion 3 has been extensively studied in previous literature, and we do not aim to quantify this measure either.
> > >
> > > We observe empirically that for image classification tasks, our inductive bias complexity often aligns with human intuition for the difficulty of a task (notion 1). However, we also find that detail-limited tasks such as Omniglot may require drastically more inductive bias than detail-rich tasks such as ImageNet, despite Omniglot being arguably intuitively easier for humans (humans can achieve roughly 5% generalization error on both tasks considering top 5 error rate for ImageNet). Similarly, as we find in our additional experiments (see our general response), combinations of simple tasks such as MNIST can lead to tasks requiring dramatically more inductive bias, which runs counter to the human intuition that solving a combination of two simple tasks must also be simple. This indicates that "inductive bias complexity" is *not* defined as human intuition for difficulty of a task (i.e. notion 1).
> > >
> > > In our experimental section, we use the term "task difficulty" as another term for "inductive bias complexity." We define "task difficulty" in this way because "inductive bias complexity" captures the amount of information necessary to generalize on a task, and thus captures the difficulty of model design. However, we emphasize that our notion of "task difficulty" *may not align with human intuition for which tasks are more difficult than others* (notion 1). Our notion of task difficulty is closer to the difficulty for a researcher to develop a well-generalizing model class (notion 2); indeed, we propose using our measure of task difficulty as a quantitative proxy for notion 2 of task difficulty to facilitate better task design. However, there are two key differences between our inductive bias complexity and notion 2: 1) our inductive bias complexity is quantifiable, whereas notion 2 is subjective and researcher-dependent, 2) our measure captures constraints on a model that may not come directly from a researcher including the standard operations available in the software used to design the model and choice of computing infrastructure.
> > >
> > > Finally, our measure of "inductive bias complexity" does not depend on the model class applied to a task (please observe that our measure of inductive bias in Eqn 11 does not depend on any quantity related to the model class actually used to train on a task). In contrast, notion 3 of task difficulty is explicitly dependent on model class. Moreover, it does not provide a useful measure of which tasks require greater care in model design; generalization error is not comparable between all tasks, and tasks with higher generalization error for a particular model class may not in general require greater inductive bias to solve. Thus, notion 3 is distinct from our notion of inductive bias complexity. Please see our discussion below for further details.
> > >
> > > In summary, we believe our notion of inductive bias complexity *is not* merely defined as the intuitive "ease of a task", however that is defined. We would be eager to hear from the reviewer how our counterintuitive findings for inductive bias complexity could be explained if our definition of inductive bias were merely tautological.

---

> > > > ### Author Response · Authors · 2022-11-24
> > > > **Thank you for your response! (2/2)**
> > > >
> > > > **Concerns on novelty of scaling results**
> > > >
> > > > We would like to emphasize that the scaling of generalization loss is *not* the equivalent to the scaling of inductive bias, and there is no obvious way to link the two types of scalings. First of all, loss scalings are typically derived as a function of the number of training samples, the number of parameters in an architecture and the intrinsic dimensionality of the data. Observe that the amount of inductive bias is *not* an input in these scaling results (typically, the results only depend on aggregate properties of an architecture such as the number of layers or parameters). Moreover, even assuming the inductive bias arising from an architecture can be fully varied, the inductive bias arising from other sources such as the random initialization of a network or the optimization method is fixed. Most importantly, these different sources of inductive bias are not directly and quantitatively comparable: knowing the scaling of loss as a function of properties of the architecture can only be used to determine how many parameters or layers in an architecture are required to generalize, for example- it *cannot* be used to quantify how much total inductive bias is required to generalize.
> > > >
> > > > By contrast, inductive bias scalings depend purely on properties of the task including the desired loss (please see Eqn 11 for the dependencies of inductive bias complexity). The inductive bias quantified here includes *all* possible sources of inductive bias, including those from the architecture, the random initialization and the optimization method. This allows us to compare how much inductive bias is required to generalize on different tasks, and thus provides a quantitative guide to the construction of tasks requiring greater inductive bias. Because scaling results for generalization loss don't consider all sources of inductive bias, they unfortunately can't be used in this way; loss scaling results can only determine the required properties of an architecture to generalize on a task *given a particular learning setup* (such as a parameterization for the architecture and an optimization method). In other words, loss scaling results only provide a guide to the construction of tasks requiring a greater number of layers or parameters *in the context of a particular architecture.*
> > > >
> > > > **Concerns on model agnosticism of framework**
> > > >
> > > > We would like to emphasize that our measure of inductive bias complexity is a property of a task, not a property of the model class applied to a task. Thus, inductive bias complexity is *by definition* model-agnostic: it does not depend on the model class actually used on a task.
> > > >
> > > > As the reviewer notes, we do indeed construct a broad and general model class within our computation of inductive bias complexity. However, this is *merely a mathematical construct* used to quantify task difficulty and has *no relationship* to the actual model classes practically applied on a task. The base model class is constructed linearly as a weighted sum of eigenfunctions of the Laplace-Beltrami operator applied to the manifold; by contrast, typical model classes for the tasks we consider are neural networks (with a much smaller dimensionality than our base model class). Of course, our measure of inductive bias complexity *does* depend on the selection of this base model class (please see Section F.5 for empirical results on how the choice of model class affects inductive bias complexity). However, the choice of base model class is *purely* a function of the properties of a task. Thus, inductive bias complexity is also purely a property of a task. We would be happy to further clarify this point in future revisions of our paper.

---

### Official Review · Reviewer_NEm3 · 2022-10-23

**Confidence:** 3
**Correctness:** 2
**Technical Novelty And Significance:** 3
**Empirical Novelty And Significance:** 3
**Recommendation:** 3

**Clarity, Quality, Novelty And Reproducibility:**

This work is at times very informal and arguments are often only vaguely outlined and prior work is used without giving much context. This made it very difficult for me to follow the arguments and verify their validity. Especially Section 4.2 is very informal and assumptions are listed on-the-go or specified after the calculations are done, making it very difficult to follow. The fact that the hypothesis space is all of a sudden restricted to only consist of linear combinations of the eigenfunctions of the Laplace-Beltrami operator is only stated after all the calculations. Some more background on technical tools such as the Laplace-Beltrami operator and its eigendecomposition would help unfamiliar readers to better understand this work. I personally found it very difficult to follow this section and still cannot really tell if it is entirely correct. I am listing my more precise questions in the following:

1. I find the data model used (i.e. x) somewhat confusing. While the data also consists of continuous features, later in the work it seems that only the discrete components of x are considered (i.e. Section 4.2). How would the results change if instead of a general manifold, the features would come from standard Euclidean space? In general, it might be helpful for the reader to maybe walk through a simpler example, say supervised learning on \mathbb{R}^{d} instead of only this very abstract setting.

2. How is the implicit bias of a model defined? This term is already abused to mean so many things in deep learning that a proper definition is needed in my opinion. What is \mathcal{K} in equation (3)? You define it as the event that the model fulfills some implicit bias, but what does that mean? In equation (3) you consider minimizing over \mathcal{K} but how can we minimize over a given, fixed event? In equation (6), \mathcal{K} is then all of a sudden chosen as the event that the hypothesis generalizes with error upper-bounded by \epsilon. How does fulfilling some implicit bias translate to generalizing with error \epsilon? In my opinion, it would anyways be simpler to define task difficulty directly by equation (6).

3. In equation (9), you divide by the volume of hypotheses that interpolate the training set.  It is not clear to me why this volume is positive under uniform weighting of hypotheses. The probability that a continuous random variable (here the training loss) takes a particular value is always zero. Without specific assumptions on the loss and the model class it is not obvious to me how equation (9) makes sense.

4. The assumptions in Theorem 1 on the function space and loss function are somewhat non-standard. For which function classes does the Lipschitz-like property apply? Which loss function satisfy the listed assumptions? What loss functions do you consider for the tasks in the numerical evaluation section, i.e. for reinforcement learning etc? Do these loss functions meet the assumptions needed?

5. The Wasserstein distance approximation seems very crude and needs further motivation in my opinion. How close must q be to uniform in order for the bound to be reasonable? It seems quite surprising to me that one can arrive at a reasonable estimate of the Wasserstein distance without making any assumptions on the two distributions, other than the fact that one is an “atomised” version of the other.

6. Why is it valid to approximate the set of interpolating solutions as a ball? If the loss is continuous (e.g. mse), then even slightly perturbing the parameters breaks interpolation to my knowledge.

7. If I understand correctly, in section 4.2 you restrict the hypothesis space to linear combinations of the Laplace-Beltrami operator over the manifold. Could you give some intuition on the size of this space? How does it look for instance if we were to consider standard Euclidean space instead of a general manifold?

8. This might be minor but how do you define a uniform distribution over an uncountable, non-compact set? Even for the model class of linear regression we are dealing with an uncountable, non-compact set of parameters (i.e. \mathbb{R}^{d}) so it is not clear to me how one would reparametrize in this case.

9. Some of the figures are not sharp on my monitor, reducing readibility.

**Strength And Weaknesses:**

**Strengths**

1. The idea to reverse roles and instead assess the difficulty of a task in a way agnostic to the model class used is very interesting. The term "inductive bias” has been used very vaguely in the deep learning literature and a theoretically grounded measure for it is a very important contribution.

2. The studied setting is very general and not many assumptions are made to arrive at a tractable metric for inductive bias. This makes this work applicable to a wide variety of problems.

**Weaknesses**

1. One of the major weaknesses of this work is its readibility. Large parts of the paper are very informally written (especially the entire section 4.2), which makes it very difficult to assess the validity of the results. I have listed my specific questions in the clarity section. I would greatly appreciate if the authors could clarify my concerns.

2. While the great level of generality is certainly a strength of this work, it is also one of its bigger weaknesses. The data model which encapsulates a wide variety of tasks is very unintuitive and not easy to follow. While the authors explain in more detail how this model captures more specific settings in the appendix, I think more time could be spent in the main text to familiarize the reader with the framework. Walking through a simpler setting (i.e. supervised learning and Euclidean space) would be very helpful here.

3. The setting of this work is extremely broad, the task considered can range from supervised learning to reinforcement learning, no assumptions on the training nor the test distribution are made, and the constructed hypothesis space also seems very general. While this is certainly a great feature of this work, it also remains unclear to me how much inductive bias can really be captured if so little assumptions are made on the data distribution. What happens for instance if instead of using clean targets on ImageNet, we study randomized targets, or varied the number of classes? What happens if we add Gaussian noise to input images and thus inflate the intrinsic dimensionality of the manifold? Do we really need to pay such a high price in terms of inductive bias? I think the results of this work would be easier to put into perspective if such settings were considered. It is not clear to me why Omniglot would require way more inductive bias than ImageNet as it is not so clear how to compare these two tasks intuitively. The authors give some insights by varying the sample size on the same task but somewhat un-intuitively, the amount of needed inductive bias does not decrease too much.

**Summary Of The Paper:**

This paper studies the amount of inductive bias needed to fit a dataset and derives a computationally feasible proxy for such a measure. In contrast to sample complexity, this metric aims to measure the difficulty of learning a given dataset, i.e. how much inductive bias needs to be present in order to achieve a certain level of generalization at a given sample size. The derived metric is largely agnostic to the model space and works for a very broad set of tasks, enabling the comparison of seemingly very remote tasks such as standard supervised learning on ImageNet and reinforcement learning.


**Summary Of The Review:**

The idea of this work is very interesting, having a notion to estimate the amount of implicit bias needed to learn a dataset is very important and could lead to some new insights. In its current version however, I find this work very difficult to read and follow, large parts of the arguments are presented rather informally and the great generality of the result with only minimal assumptions on the data distribution seems somewhat surprising. More empirical evaluations for intuitively comparable setting would help the understanding as well. In summary, I have to recommend rejection of this work as in its current form, I find myself unable to confirm its validity, both theoretically and empirically. It is however very much possible that I have misunderstood parts of this work and I am happy to change my score if the authors can answer my questions.

---

> ### Author Response · Authors · 2022-11-18
> **Authors' Response to Reviewer**
>
> **Comments on volume of interpolating hypothesis space**
>
> We thank the reviewer for this valuable comment. In response, we have restated the final version of Theorem 1 in terms of conditional probability instead of volumes. We believe this circumvents the issue where the volume of $\Theta_q$ may be $0$.
>
> **Concerns on Wasserstein distance approximation**
>
> We agree with the reviewer that our Wasserstein distance approximation could be further validated. In response, we have conducted additional experiments to validate our approximation in our revised Appendix C. Specifically, we empirically compute the Wasserstein distance between a uniform distribution on a sphere and an empirical distribution drawn i.i.d. from the uniform distribution. We perform this computation for varying dimensionalities m and varying numbers of samples n. We find that our theoretically predicted scalings of Wasserstein distance, namely that the distance scales as $O(n^{-1/m})$, is reasonable.
>
> **Intuition for size of base hypothesis space**
>
> To gain some intuition for the size of the base hypothesis space, it can be useful to consider a simple Euclidean manifold. Suppose inputs lie on a unit hypercube in $m$ dimensions with periodic boundary conditions (in the sense that the top of each hypercube maps to the bottom in each dimension). In this case, the eigenfunctions of the Laplace-Beltrami operator are simply Fourier basis functions (complex exponentials). The hypothesis space is expressed as a Fourier series, and the dimensionality of the space is the number of basis functions in the Fourier series.
>
> Intuitively, what determines the number of basis functions? Recall that in $m$ dimensions, each Fourier basis function is characterized by a frequency vector in $m$ dimensions. The number of possible values for each dimension of the frequency vector is governed by the maximum frequency relevant to the task. Thus, the number of basis functions scales as the number of relevant frequencies raised to the $m$; adding one more dimension to the task multiplies the number of basis functions by the number of relevant frequencies.
>
> Since the number of basis functions scales exponentially with the number of data dimensions $m$, the hypothesis space can be very high dimensional even with a relatively small number of data dimensions. This also explains the large numbers we find for inductive bias complexity in Section 5; since the hypothesis space is so high dimensional, specifying a well-generalizing subset in this high dimensional space can require large amounts of information.
>
> Please see our revised Appendix G for further discussion on why inductive bias complexity scales exponentially with data dimensionality.
>
> **How to define uniform distribution over an uncountable, non-compact set?**
>
> We assume that our prior distribution over the base hypothesis space is uniform. However, as the reviewer notes, this can be problematic when the hypothesis space is non-compact since in these cases it may not be possible to define a uniform distribution. However, note that in practice, our parameterization of the hypothesis space in Section 4.3 produces a *bounded* hypothesis space. It is possible to define a uniform distribution over this bounded space.
>
> As we note in Appendix B, in general, many prior distributions over hypotheses can be converted to uniform distributions by the appropriate choice of reparameterization. Typically, if the original prior distribution has a bounded support, then so will the support of the uniform distribution over the reparameterized parameters.

---

### Official Review · Reviewer_FUck · 2022-10-25

**Confidence:** 3
**Correctness:** 2
**Technical Novelty And Significance:** 3
**Empirical Novelty And Significance:** 2
**Recommendation:** 3

**Clarity, Quality, Novelty And Reproducibility:**

In general the paper is hard to follow and unclear in some places, especially in Section 4.2. Some of the problems come from trying to accommodate many learning settings under one umbrella. As this is the first step towards defining inductive bias complexity, maybe one just needs to consider only the standard supervised classification setting first.

**Strength And Weaknesses:**

#### **Strengths**

* This submission focuses on an important problem that was not addressed much to my best knowledge.
* The proposed notion of inductive bias complexity is interesting.

#### **Weakness 1: limitations**

1. The proposed task difficulty depends on a large hypothesis set with a uniform prior on it. It is unclear how to select this hypothesis set and how it affects the findings.
2. A uniform distribution might not exist for some of such large hypothesis sets.
3. Even when a uniform prior can exist, it is still an arbitrary choice, as it disregards the geometry of the space. Maybe a more appropriate choice would be the Jeffreys prior.
4. In the meta-learning setting this hypothesis set needs to be a set of mappings from datasets to functions. It is unclear how even distributions are defined on such spaces. Could you please elaborate on this?
5. The proposed task depends only on the set of interpolating hypotheses. As mentioned in the paper, there might be well-generalizing but non-interpolating hypotheses.

#### **Weakness 2: unrealistic assumptions and simplifications**
1. In Thm 1, “Suppose that the loss function is invariant to shifts $L(y_1 + y, y_2 + y) = L(y_1, y_2)$ for all $y_1, y_2, y$.”:  This assumes that there is an addition operation defined on the action space. How does this hold in the settings of supervised classification or meta-learning for few-shot classification?
2. The assumption of Eq. (8) seems too strong. Can you give examples of large hypothesis sets for which this assumption holds?
3. In Sec 4.2, the data distribution is approximated as a mixture of uniform distributions on disjoint spheres. In particular, in a supervised classification setting, it is assumed that examples of the same class have a uniform distribution on a sphere.

#### **Weakness 3: It is hard to validate the experiment findings.**
1. Since there are no baselines presented and the proposed approach does many simplifications, it is hard to judge the validity of the experimental findings.
2. I recommend designing some sanity checks for validating the results.  For example, how does the difficulty of union of two related (or independent) tasks compare to those of the individual tasks?
3. In Figure 3(a), ImageNet training data is hypothetically increased to the size more than $10^{17}$, but we see that the approximated task difficulty remains roughly the same. It seems that with so many training examples, the set of interpolating solutions should contain only the ground-truth function (or extremely similar ones), therefore making the task difficulty close to zero. This entails that we should at least see a large drop in the approximated task difficulty when the number of examples is so large.


**One related paper:** Achille et al. [1] also define information-theoretic model-agnostic task complexities.

#### **References**

[1] Achille, A., Paolini, G., Mbeng, G. and Soatto, S., 2021. The information complexity of learning tasks, their structure and their distance. Information and Inference: A Journal of the IMA, 10(1), pp.51-72.

**Summary Of The Paper:**

This submission tackles the problem of measuring inherent difficulty of a learning task, independent of a learning algorithm and model class. Formally, a learning task is defined as a training set of instances and corresponding actions sampled from a training distribution. The goal of a learner is to generalize well to instances sampled from a possibly related test distribution. It is assumed that there is a ground-truth mapping from instances to actions. It is also assumed that one has access to a large hypothesis set (different from the model class) with a uniform prior on it, that includes the ground truth mapping. The authors show that common learning settings, such as supervised classification, reinforcement learning, meta-learning for few-shot classification, and meta-reinforcement learning can be formulated in this framework.

Within this framework, the authors introduce a measure of task difficulty that quantifies the minimum “amount of inductive biases" needed to generalize well. This boils down to computing the fraction of interpolating hypotheses that have a desired level of performance on the test distribution. As it is hard to compute this quantity exactly, the rest of the paper makes a series of simpliciations and assumptions in order to compute an upper bound for this quantity. The final section of the paper computes this upper bound for a few standard learning tasks and examines the trends of it with different aspects of learning tasks.


**Summary Of The Review:**

In summary, while I like the problem and the perspective of this submission, I find the paper not ready for publication.

---

> ### Author Response · Authors · 2022-11-18
> **Authors' Response to Reviewer (1/?)**
>
> **How does the choice of hypothesis space affect findings?**
>
> As the reviewer notes, we select a large hypothesis set with a uniform prior placed on it. As we motivate and describe in Section 4.3, the hypothesis space is constructed to include all bandlimited functions below a certain task-relevant frequency threshold. We believe this is a reasonable choice since we wish to select a hypothesis set which is sufficiently broad to include any model class that may reasonably be applied to a task, while at the same time being bounded and finite-dimensional.
>
> However, different choices of hypothesis set can yield different values for inductive bias. One way to parametrically vary the hypothesis space is to vary the relevant spatial resolution of the task $\delta$. We conduct additional experiments varying $\delta$ for ImageNet and observing changes in inductive bias complexity (see Appendix F for full results). We find that the choice of $\delta$ is indeed critical to determining the scale of inductive bias complexity; specifically, varying $\delta$ from 30 to 120 can vary the scale of inductive bias complexity by nearly 30 orders of magnitude. Thus, selecting the task-relevant input resolution to consider when constructing a hypothesis space is crucial. For all our experiments, we have selected the input resolution in a clear and consistent way as described in Appendix D; thus, we believe the resulting task difficulties can be meaningfully compared even if there is some error in the selection of $\delta$ for each task.
>
> Regarding the choice of a uniform prior over the hypothesis space, we note that selecting a uniform prior is in alignment with the principle of maximum entropy. In our quantification of inductive bias complexity, we aim to minimize the assumptions we make about the structure of the model class, so selecting a uniform prior over all hypotheses seems appropriate. Indeed, there may be other priors that may be more appropriate in specific settings. However, the uniform prior is simple and generally applicable.
>
> Regarding the existence of a uniform prior, as the reviewer observes, it is true that a uniform prior may not exist for all hypothesis sets, particularly those over uncountable, non-compact sets. However, our parameterization of hypothesis in Section 4.3 produces a *bounded* hypothesis set over which it is possible to define a uniform prior.
>
> Regarding the use of the Jeffreys prior over the hypothesis space, we note that the Jeffreys prior is appropriate when it provides a probability distribution over a parameter that itself parameterizes another probability distribution. However, in our case, the hypothesis may not necessarily parameterize probability distributions, instead parameterizing arbitrary functions. Thus, a Jeffreys prior is not applicable to our setting, and a uniform prior may be a more appropriate choice.
>
> Regarding how the hypothesis space is defined for meta-learning problems, first note that for meta-learning problems, each hypothesis corresponds to a mapping from datasets to classifiers. Each of these classifiers maps from a finite dimensional input space to a finite dimensional output space; these classifiers are parameterized by a finite number of parameters using the same approach used to parameterize general hypothesis spaces (using a linear combination of a finite set of basis functions). Since the classifiers can be described by a finite number of parameters, the hypotheses of the original meta-learning problem then correspond to a mapping from the finite dimensional space of datasets to the finite dimensional space of classifiers. We then may apply the same hypothesis parameterization described in Section 4.3 to the meta-learning problem. See Appendix D.2 for further details on how to define the hypothesis space for meta-learning problems.

---

> > ### Author Response · Authors · 2022-11-18
> > **Authors' Response to Reviewer (2/?)**
> >
> > **Concerns on only considering interpolating hypotheses**
> >
> > As the reviewer notes, one limitation of our work is that we only consider interpolating hypotheses (in other words, we only consider training error $\epsilon=0$ in our revised Equation 3). Although we will leave a formal extension of our results to non-interpolating hypotheses as a future work, we briefly outline how our results might be extended:
> >
> > First, we may provide an analogous result to Theorem 1 in the non-interpolating case by arguing that non-interpolating hypotheses may generalize well as long as they are within a smaller radius of the true hypotheses (relative to the radius for interpolating hypotheses). Specifically, we may expect the more general expression for the radius to be $\frac{\varepsilon - \epsilon}{L_\mathcal{L}L_fW(p,q)}$.
> >
> > Next, in order to quantify probabilities in the hypothesis space and find a practical estimate for inductive bias complexity, we must quantify the size of the hypothesis space that fits the training data up to error $\epsilon$. Intuitively, we may expect it to be a region around the interpolating hypothesis space with dimensionality equal to that of the base hypothesis space. We may expect the size of this region to scale polynomially with $\epsilon$. As with the interpolating case, we can then estimate the fraction of the near-interpolating hypothesis space that is close enough to the true hypothesis to find a generalized, practically-computable expression for task difficulty.
> >
> > **Additional reference**
> >
> > We thank the reviewer for pointing out this reference. We have added a discussion of it in our revised Section 2.

---

### Official Review · Reviewer_eajX · 2022-10-26

**Confidence:** 3
**Correctness:** 3
**Technical Novelty And Significance:** 3
**Empirical Novelty And Significance:** 2
**Recommendation:** 8

**Clarity, Quality, Novelty And Reproducibility:**

Clarity is very good

Quality is good

Novelty is very good

Reproducibility could be made more clear. Some of the experimental details could be discussed in the main text as well.

**Strength And Weaknesses:**

# Strength
1. I find the paper written in a very clear manner. I have enjoyed a lot reading the paper!
2. The notion of "inductive bias complexity" is very novel to my knowledge, and is also very interesting.
3. Both theory and experiments are provided for understanding the proposed inductive bias complexity, which form the work in a rather mature shape.

# Questions
I think the work is very interesting and is worthy getting more attention from the community. I have the following questions/suggestions:

1. I think Appendix C is crucial for understand the empirical results. Perhaps it is worthy to summarizing some of the important quantities (e.g., $m, \\delta, \\epsilon$), as well as the methodology for estimating them, and adding them into main text?

2. I feel the "inductive bias" could have been formally defined. Based on my understanding, it refers to a restricted set of models (as a subset of the full hypothesis class) that can be outputted by some fixed procedures/code (or algorithms, with some fixed or tunable hyperparameters?).

3. It might also be worthy to discussing the sources of randomness. To my understanding the randomness is from the randomly drawing of the training samples (correct me if not). How does the framework account the randomness in the learning algorithm? For example, people believe that the random nature of SGD encodes certain inductive bias.

4. While I understand that the current theory might be in their ultimate sharp version, I would like to see some discussion on why the inductive bias complexity has an exponential dependence on the data dimension. Would you say this exponential dependence is fundamental and un-improvable or just technical? Could you discuss some intuition on the exponential dependence?

5. Some discussion in Section 5.2 is not very clear. In particular, how do you use test data in computing the inductive bias complexity? Do you treat the test error as the $\\epsilon$ in your formula?

6. I am curious how useful is the inductive bias complexity in practice, even though I think it is already very interesting in terms of theory. Let us say we are given with a training set and a task, what else do we need to compute the inductive bias complexity? Or how do we compare the inductive bias complexity to that of other dataset+task? What implication can we get, if we obtain some bounds/estimations on the inductive bias complexity?



**Summary Of The Paper:**

This work proposes a novel notion called "inductive bias complexity". Just as sample complexity captures the number of samples required for learning a problem (upto a desired error) given a fixed inductive bias, the "inductive bias complexity" refers to the minimum amount of "inductive bias" required for learning a problem (upto a desired error) given a fixed number of training data. This notion is interesting because it, to some extent, explains the amount of prior knowledge (or handcrafted arts) one needs to set in order to solve a problem with the given amount of data.

After introducing the notion of "inductive bias complexity", this work further provides a bound on it based on a set of conditions, which are somewhat strong but are also quite standard in learning theory. Finally, some empirical results are provided to verify the theoretical predictions.

**Summary Of The Review:**

At this point I am very positive to this work. The authors reply on my questions can help me better understand the work.

---

> ### Author Response · Authors · 2022-11-18
> **Authors' Response to Reviewer (1/?)**
>
> **Summarizing estimation of main quantities in main text**
>
> We thank the reviewer for this valuable suggestion. In response, we have expanded our description of our estimation approach for task parameters in Section 5 of the main text (leaving the full details in Appendix D).
>
> **Questions on sources of randomness**
>
> We would first like to clarify that there are two important categories of randomness: randomness in the generation of the dataset and randomness in the learning algorithm. We discuss them one by one:
>
> Randomness in the generation of the dataset arises from the random sampling of training data from the underlying data distribution. Since the inductive bias complexity is a property inherent to each dataset, different random samplings may in general yield different inductive bias complexities. For instance, a random sampling of training data that is closer to the underlying distribution (in terms of Wasserstein distance) may have lower inductive bias complexity compared to a random sampling that is farther away.
>
> Next, we consider randomness in the learning algorithm (such as random sampling in SGD). Note that our measure of inductive bias complexity is a property inherent to the *dataset* on which a learning algorithm is trained and the generalization error achieved by the learning algorithm, not the learning algorithm itself. Thus, the extent to which randomness in learning algorithms encodes inductive bias is determined by how much the randomness improves the generalization performance of the learning algorithm. The observation that stochasticity in SGD improves generalization then indicates that the randomness encodes some inductive bias.
>
> Relatedly, we may consider a case where the training data is noisy in the sense that for each training instance, the optimal action may be misspecified (this corresponds to label noise in the case of supervised classification). We may view this as noise in the learning algorithm where noise is applied to the noiseless training data before the remainder of the learning algorithm is used. In this setting, the set of "interpolating" hypotheses (that interpolate the noisy data) may *not* include the true hypotheses since these hypotheses only interpolate samples from a noisy version of the true hypothesis. Thus, our assumption that relevant hypotheses interpolate the training data may not apply.
>
> Nevertheless, we incorporate this setting into our framework as follows: instead of considering only hypotheses that perfectly interpolate the noisy data, we may instead consider hypotheses that achieve up to a fixed error rate on the training dataset. This fixed error rate may be selected such that the true hypothesis achieves this error rate (or lower) on the training set. Next, to compute inductive bias complexity, we may compute the fraction of hypotheses that generalize well among the hypotheses that fit the training dataset up to the fixed error rate. Just as in the interpolating case, a smaller fraction of generalizing hypotheses corresponds to a larger inductive bias.
>
> Please see our revised Appendix G for additional discussion on how to extend our framework to non-interpolating hypotheses.

---

> > ### Author Response · Authors · 2022-11-18
> > **Authors' Response to Reviewer (2/?)**
> >
> > **Why does inductive bias exponentially depend on data dimension?**
> >
> > Intuitively, inductive bias scales exponentially with data dimension because inductive bias scales with the dimensionality of the hypothesis space, and the dimensionality of the hypothesis space scales exponentially with data dimension.
> >
> > First, we consider the intuition for why inductive bias scales with the dimensionality of the hypothesis space: recall that generalizing requires specifying a specific set of well-generalizing hypotheses in the hypothesis space. The amount of information needed to specify a point in an $m$ dimensional space scales with $m$ since it is simply the number of coordinates of the point. Thus, assuming well-generalizing hypotheses are concentrated in a region around a point, the amount of information required to specify the region also scales with the dimensionality of the hypothesis space.
> >
> > Next, we consider the intuition for why the dimensionality of the hypothesis space scales with data dimensionality: consider a very simple "grid-based" method of constructing hypotheses in which the input manifold is divided into equally sized hypercubes that tile the entire manifold. A hypothesis consists of a mapping between hypercubes and outputs. Note that the dimensionality of each hypothesis scales with the number of hypercubes since each hypothesis can be specified by its output at each hypercube. The number of hypercubes scales exponentially with the dimensionality of the input manifold; thus the hypothesis space dimensionality also scales exponentially with the dimensionality of the input manifold.
> >
> > We view this exponential dependence as fundamental: generalizing over more dimensions of variation significantly expands the hypothesis space regardless of how we parameterize hypotheses, and thus dramatically increases inductive bias complexity.
> >
> > We have clarified these points in our revised Appendix G.
> >
> > **Questions on use of test data in inductive bias computation**
> >
> > To compute the inductive bias of a specific model, we first compute its test set error rate (*not* its training set error rate). We then use this as the desired error rate $\varepsilon$ in our inductive bias formula in Equation 11. Importantly, we do *not* directly use the test set data to compute the inductive bias complexity. Instead, the test set data is only used in combination with a trained model to compute a test set error rate.
> >
> > We have clarified this point in our revised Section 5.2.

---

### Author Response · Authors · 2022-11-18
**General Response (1/?)**

We thank all reviewers for their valuable comments and suggestions. We are glad to hear that the reviewers find our work, which generates the first theoretical measure of inductive bias complexity, novel and important. In response to their concerns, we have made major revisions to our submission and have conducted additional experiments including task difficulty computations for additional task variations and empirical validations of our theoretical assumptions. We believe these changes can address all concerns raised by the reviewers and we would be happy to answer any remaining questions.

**Additional variations of tasks to better validate experimental findings**

We thank the reviewers for their suggestions on additional task variations to better validate our task difficulty measurements (in addition to the ones we have already included in Section 5.3). We have computed task difficulties for these additional task variations (the new results are included in our revised appendix F).

First, we compute task difficulties for combinations of tasks. The task combinations are constructed as follows: given task 1 that maps instances x_1 to actions y_1 and task 2 mapping instances x_2 to actions y_2, we define the combination of task 1 and task 2 as the one mapping instances (x_1, x_2) to actions (y_1, y_2). As argued in our revised Appendix F, we show that the difficulty of the combined task can be bounded based on the difficulties of versions of task 1 and task 2 where *distractor* inputs from the *other* task are added.

We compute task difficulties for pairwise combinations of MNIST, SVHN, and CIFAR-10 (which have individual task difficulties (in bits) of $1 \times 10^{16}$, $1 \times 10^{31}$, $3 \times 10^{32}$). As found in the table below, combined task difficulties are significantly greater than the difficulties of individual tasks. As a very rough rule of thumb, the combined task difficulty is approximately the product of the individual task difficulties. This makes sense since the task difficulty scales exponentially with the intrinsic dimension of the data and combining together two tasks *adds* together the intrinsic dimensionality of the two tasks (and thus multiplies their task difficulties).


**Tab R1: Combined task difficulty bounds (in bits) for combinations of image classification tasks. Difficulties are reported as "(lower bound, upper bound)."**
| Dataset | MNIST | SVHN | CIFAR-10 |
|:---|:---:|:---:|:---:|
| **MNIST**    | ($1 \times 10^{26}$, $3 \times 10^{26}$) | ($6 \times 10^{39}$, $5 \times 10^{45}$) | ($3 \times 10^{42}$, $8 \times 10^{44}$)  |
| **SVHN**     | -     | ($4 \times 10^{54}$, $8 \times 10^{54}$)  | ($4 \times 10^{50}$, $4 \times 10^{61}$) |
| **CIFAR-10** | -   | -   | ($2 \times 10^{55}$, $3 \times 10^{55}$)  |


Why does combining together two simple tasks like MNIST result in a much more difficult task? This is because separating the two parts of the combined task requires a lot of inductive bias. Without this inductive bias, solving a combination of two tasks looks like solving a single task with a higher dimensional input, which correspondingly requires a much greater inductive bias. This also suggests how practical model classes like neural networks may be able to provide such vast amounts of inductive bias to a task: namely, by breaking down the input into lower-dimensional components.

We additionally compute task difficulties for the following variations of ImageNet: 1) mapping all images to an output of zero, 2) mapping all images to a random target function, 3) varying the number of classes, 4) varying the spatial resolution of the task $\delta$.

1. In the zero-output setting, the task difficulty depends critically on how the hypothesis space is constructed. If the hypothesis space is constructed based on the task-relevant resolution as described in Section 4.3, then the hypothesis space would only contain a single hypothesis (the zero hypothesis), and the task difficulty would be $0$. However, if we construct the hypothesis space the same way as for the original ImageNet, then the task difficulty is the same as for the original ImageNet. Intuitively, this is because the hypothesis space does not favor the zero hypothesis over the hypothesis of the original ImageNet: both require equal amounts of information to specify.

---

> ### Author Response · Authors · 2022-11-18
> **General Response (2/?)**
>
> 2. In the setting where ImageNet inputs are mapped to a random target function chosen from the original hypothesis space, the task difficulty is again the same as for ImageNet. The intuition is the same as for the zero-output setting: the hypothesis space does not favor the original ImageNet hypothesis over a random one from the hypothesis space.
>
> 3. We find that as the number of classes is varied on ImageNet from $10$ to $1000$, the task difficulty increases by roughly 5 orders of magnitude (please see Figure 5a in our revision). This is primarily due to the increased number of manifolds over which hypotheses are defined when there are more classes: the hypothesis space increases with more classes. Thus, adding more classes to a task can be a moderately useful way of increasing its difficulty (although not as powerful as increasing its intrinsic dimensionality).
>
> 4. We find that as the spatial resolution of ImageNet is varied from $30$ to $120$, the task difficulty drops by nearly 30 orders of magnitude. Recall that the spatial resolution controls the hypothesis space of a task; decreasing spatial resolution increases the size of the base hypothesis space. This results in much higher difficulty in specifying regions in the hypothesis space. These results emphasize the critical effect of the selection of base hypothesis space on task difficulty.
>
> Please see our revised Appendix F for full results and discussion.
>
> **Concerns on clarity (overall and Section 4)**
>
> We thank the reviewers for their valuable suggestions on improving the clarity of Section 4. In response, we have significantly revised and restructured the section, including adding a formal definition of inductive bias complexity in our new section 4.1, considering only a supervised classification setting in Euclidean space for our approximations in our new section 4.3, explicitly summarizing all assumptions made in our empirical approximations at the beginning of section 4.3, providing additional background on the Laplace-Beltrami operator and improving our notation to be more consistent and clear.
>
> We also make a number of additional changes to improve the clarity of the paper overall:
>
> - To clarify the parameters of a task controlling its inductive bias complexity, we have changed all parameters inherent to a task to be lowercase.
>
> - To clarify that the loss function may depend on instances $x$, we have expressed the loss function as $\mathcal{L}(f(x), x)$.
>
> - We modify the final form of Theorem 1 in terms of conditional probabilities instead of expressing the inductive bias complexity in terms of volumes in the hypothesis space. We hope this clarifies the intuition for the form of inductive bias complexity as well as addresses any potential technical issues with quantifying volumes in the hypothesis space as raised by Reviewer **NEm3**.
>
> - We have modified Figure 1 to be more clear.
>
> - We have added a new discussion section (Appendix G) in which we discuss and clarify important points related to the interpretation of our theoretical and empirical results.
>
> - As suggested by Reviewer **eajX**, we have expanded our description of our estimation approach for task parameters in Section 5 of the main text (leaving the full details in Appendix D).
>
> - We have added a new Figure 6 to better illustrate our construction of hypothesis spaces in Section 4.3.

---

> > ### Author Response · Authors · 2022-11-18
> > **General Response (3/?)**
> >
> > **Concerns on unrealistic theoretical assumptions**
> >
> > We agree with the reviewers that several assumptions and simplifications are required to arrive at our final task difficulty approximation in Equation 14. However, we believe that these assumptions are reasonable, and are justified in that they enable us to produce the first quantitative measure of inductive bias complexity.
> >
> > We justify our theoretical assumptions as follows, and we have edited our submission to include this discussion (please see our revised Appendix C and associated pointers in the main text):
> >
> > 1. Assumption of shift invariance of the loss function in Theorem 1: In our revision, we have been able to weaken the shift-invariance assumption. Please also note our modified notation.
> >
> > In many supervised classification and regression settings, the shift invariance assumption is satisfied by a squared error loss function. For clarity, in supervised regression, we denote inputs as $x$ and outputs as $y$. A squared error loss function is constructed as $\mathcal{L}(y, x) = ||y - f*(x)||_2^2$ which is shift invariant since $\mathcal{L}(y + f^*(x_1) - f^*(x_2), x_1) = ||y + f^*(x_1) - f^*(x_2) - f^*(x_1)||_2^2 = ||y-f^*(x_2)||_2^2 = \mathcal{L}(y, x_2)$.
> >
> > Next, in a few-shot meta-learning setting, there are reasonably broad conditions under which shift-invariance holds:  Instances correspond to datasets and actions correspond to functions mapping from inputs to the outputs of the inner loop task. For clarity, we define the inner task inputs and outputs as $i$ and $o$; the goal is to learn a mapping from datasets $x$ to functions $y$ that map $o=y(i)$.
> >  Furthermore, suppose a loss function $L(o, i)$ is defined for the inner loop task.
> >  Then, suppose the meta-loss function is defined as:
> >
> >  $\mathcal{L}(y, x) = \mathbb{E}_{i \sim x}[L(y(i), i)]$ .
> >
> >  We assume that functions $y$ are additive in the following sense: for any $y_1, y_2, x$, $(y_1 + y_2)(x) = y_1(x) + y_2(x)$. Finally, we assume a generalized version of shift-invariance for the inner loss function $L$:
> >
> > $\mathbb{E}_{i \sim x_2}[L(y(i), i))]$
> >
> >  $=\mathbb{E}_{i \sim x_1}[L(y(i) + f^*(x_2)(i) - f^*(x_1)(i), i)]$
> >
> > This then implies shift-invariance for the meta-loss $\mathcal{L}$:
> >
> > $\mathcal{L}(y, x_2) = \mathcal{L}(y + f^*(x_2) - f^*(x_1), x_1)$
> >
> > As the reviewers note, the shift invariance assumption implies addition is defined over the action space. We believe that such additivity is reasonable for most learning problems. For instance, in classification settings, action spaces typically correspond to a vector of logits over all classes; these action spaces are Euclidean and actions can be added. Similarly, in reinforcement learning, both continuous and discrete action spaces are often additive, with the logits of a probability distribution over possible actions outputted in the discrete case. Moreover, shift-invariance is a property of the commonly used squared error loss, and indeed any loss function that has the form $\mathcal{L}(y, x) = G(y - f^*(x))$ for some function $G$.
> >
> > At the same time, we note that many loss functions may not satisfy shift invariance. For example the error rate (computed as the fraction of incorrectly classified points in supervised classification), is neither differentiable nor  shift-invariant. Thus, during training, differentiable proxy loss functions such as squared error or cross-entropy loss are often used, to similar effect. In other words, shift-invariant proxies can exist for non-shift-invariant losses. In reinforcement learning, loss functions may correspond to a Q function computing the cumulative negative reward over an episode conditioned on taking a particular action. In this case, loss functions often cannot be written in closed form, making it difficult to show shift invariance for these functions even if they are actually shift invariant.
> >
> > For these reasons, we believe shift-invariance is a useful approximation that allows us to theoretically prove properties of task difficulty. We further note that the result of Theorem 1, which we used shift-invariance to prove, may hold even for loss functions without without shift invariance:
> >
> > $\mathcal{L}(f(x_1;\theta), x_1) - \mathcal{L}(f(x_2;\theta),x_2) \le L_\mathcal{L}L_f d(x_1,x_2) d(\theta,\theta^*)$
> >
> > Intuitively, it states that when parameters are near their optimal values, small changes in inputs $x$ yield only small changes in the loss, and this feels like a very simple and general requirement.

---

> > > ### Author Response · Authors · 2022-11-18
> > > **General Response (4/?)**
> > >
> > > 2. The second order condition in Equation 5 assumes that changes in the model $f$ can be bounded when the inputs and parameters of $f$ change a small amount. First, observe that if $f$ is twice differentiable, then the condition always holds for some choice of $L_f$ assuming $x$ and $\theta$ are defined on a bounded domain. As a concrete example, consider the case of image classification with a Tanh activated neural network where each input pixel is bounded in range $[0, 1]$ and network parameters are bounded by a large radius (the bounded parameter assumption is reasonable since practically speaking, trained neural network parameters can be bounded by some radius depending on the amount of training).
> > >
> > > As an example of a function that does not satisfy the condition, note that neural networks with the ReLU activation function may not satisfy Equation 8. However, it has been observed that ReLU networks behave like neural networks with smooth, polynomial activation functions (Poggio et al., 2018); thus, it may be reasonable to apply our assumption even to ReLU networks.
> > >
> > > Poggio, Tomaso, et al. "Theory of deep learning III: explaining the non-overfitting puzzle." arXiv preprint arXiv:1801.00173 (2018).
> > >
> > > 3. In Section 4.2, in order to arrive at a computable expression for task difficulty in the supervised classification setting, we assume that the data distribution for each class is a uniform distribution on a sphere. We believe this assumption captures some key properties of actual data distributions in common image classification tasks: 1) bounded domain, 2) similar density over all points, 3) non-overlapping distributions for different classes. We do not claim that image classification tasks *actually* follow a spherical distribution, but we use this as a model to be able to quantify task difficulty.
> > >
> > > 4. Our approximation of the Wasserstein distance in equation 8 specifically estimates the Wasserstein distance between a uniform distribution over a sphere and the empirical distribution of a sample of points drawn from the distribution. By estimating the optimal transport map as a tiling of hypercubes on the sphere, we can approximate this distance as $O(r n^{-1/m})$, where $r$ is the radius of the sphere, $n$ is the number of points and $m$ is the dimensionality of the sphere. As argued in Section 4.3, we may expect this scaling to be valid for large $n$.
> > >
> > > To validate this approximation, in our revised Appendix C, we conduct experiments empirically computing Wasserstein distances on a sphere. We find that experimentally, Wasserstein distances follow our theoretically predicted trends. In our revised Section 4.3 in the main text, we point to this Appendix section.
> > >
> > > 5. Regarding our approximation of the interpolating hypothesis space, we first note that the dimensionality of the interpolating hypothesis is *lower* than the full dimensionality of the hypothesis space. As the reviewer notes, taking an interpolating hypothesis and slightly perturbing it in an *arbitrary* direction will likely break interpolation. By contrast, we expect that there exists some subspace in which perturbations along the subspace maintain interpolation (i.e. there is a continuous manifold of interpolating hypotheses). This is analogous to how in neural networks, local minima can often be in *flat* regions in which perturbing parameters along those directions do not significantly affect the loss (Keskar et al. 2017)
> > >
> > > We approximate the interpolating hypothesis space as a disk in the overall hypothesis space (for clarity, we have revised our terminology from "ball" to "disk"); this disk has lower dimensionality than the overall hypothesis space but is embedded in a higher dimensional space. The disk assumption preserves key properties that we may reasonably expect of interpolating hypothesis spaces, namely 1) low-dimensionality and 2) boundedness, while being analytically tractable.
> > >
> > > Finally, we emphasize that the disk approximation is made to approximate the volume of the interpolating hypothesis space. The detailed shape of the interpolating hypothesis space is not itself central to our result, but rather the scaling of its volume is more important. We believe the disk assumption captures the key scalings: the dependence of the volume on $b$ and $d$.
> > >
> > > We have clarified this approximation in our revised Section 4.3.
> > >
> > > Keskar, Nitish Shirish, et al. "On large-batch training for deep learning: Generalization gap and sharp minima." ICLR 2017.

---

> > > > ### Author Response · Authors · 2022-11-18
> > > > **General Response (5/?)**
> > > >
> > > > **Formal definition of inductive bias**
> > > >
> > > > We thank the reviewers for their valuable comments regarding our definition of inductive bias. In response, we have added a new Section 4.1 in our revision that formally defines inductive bias complexity based on the probability that a random hypothesis that fits the training data also generalizes well below some error threshold. Compared to our previous definition, we more directly define inductive bias complexity without defining specialized probabilistic events in the hypothesis space. We have also referred to this formal definition in the introduction and provided a more intuitive explanation of inductive bias in the introduction itself.
> > > >
> > > > We hope this clarifies, simplifies, and provides more rigor to our notion of inductive bias complexity.
> > > >
> > > > **Practical significance of inductive bias complexity measure**
> > > >
> > > > We propose using our inductive bias complexity measure as a guide to develop tasks requiring models containing more inductive biases. Given a task with higher inductive bias complexity than another as found by our measure, we may expect that further architectural and algorithmic innovations may be required to solve the task. So far in the field, benchmarks have often been designed based on to resemble tasks in the real world or based on researchers' intuitions for which tasks may be out of reach of current models (and thus may spur development of new models). Our measure provides the first theoretically-grounded and *quantitative* guide to task design.
> > > >
> > > > Moreover, our measure does not merely allow researchers to quantitatively confirm their intuitions for which tasks are more challenging than others; it can provide surprising new insights into which tasks require more inductive bias (and thus likely more effort in model design) than others. As an example, we found that few-shot meta-learning on the Omniglot dataset requires many orders of magnitude more inductive bias than classifying Imagenet despite the apparent simplicity of Omniglot. This suggests that greater effort must be placed on algorithm and architecture design for Omniglot relative to Imagenet. Indeed, while gradient descent combined with an appropriate deep neural network architecture can be sufficient for ImageNet, generalizing on Omniglot typically requires specialized meta-learning procedures.
> > > >
> > > > As a specific example of a proposal to design more challenging tasks, in our revision, we have suggested that adding Gaussian (or other types of) noise to task inputs is a simple and general way of increasing the intrinsic dimensionality of a task, and thereby drastically increasing the inductive biases required to generalize it. For instance, it is well-known that in reinforcement learning, adding noise to observations can convert a fully-observed Markov decision process to a partially-observed one and make it significantly more challenging to solve. In supervised image classification, classifying images with adversarial perturbations is known to be significantly more challenging to generalize on than standard image classification.
> > > >
> > > > Please see our discussion section for further comments on the practical importance of measuring inductive bias complexity.

---

> > > > > ### Author Response · Authors · 2022-11-18
> > > > > **General Response (6/?)**
> > > > >
> > > > > **Why does inductive bias complexity change so little with varying amounts of training data?**
> > > > >
> > > > > We agree with the reviewers that it is intuitively quite surprising that inductive bias complexity varies so little with the amount of training data (see Section 5.3). In particular, we find that for ImageNet, even when the number of training samples is increased from $10^6$ to beyond $10^{17}$, the scale of inductive bias complexity remains around $10^{41}$ bits. We may interpret this as meaning that training points provide relatively information for generalization relative to inductive biases.
> > > > >
> > > > > **Why do training points provide so little information?** Intuitively, it is because in the absence of strong constraints on the hypothesis space, training points only provide information about the function we aim to approximate in a *local* neighborhood around each training point. By contrast, inductive biases can provide more *global* constraints on the hypothesis space than can dramatically reduce the size of the hypothesis space and allow for generalization. Without strong inductive biases, training points need to cover large regions of the input manifold to allow for generalization. With high dimensional manifolds, this can require very large numbers of training points.
> > > > >
> > > > > Through simple scaling arguments (such as those cited by Reviewer **Du5k**), we can estimate the number of training samples we need to generalize on ImageNet in the absence of strong inductive biases. We estimate the intrinsic dimensionality of ImageNet to be $48$ and the task-relevant resolution $\delta$ of ImageNet to be $65$. We can then estimate the number of training points needed to generalize as the number such that any region on the manifold is within radius $\delta$ of a training point. With $n$ training points, we can expect to cover a volume of about $65^{48}n$ on the manifold. Given a manifold of volume about $(255 \times 224 \times \sqrt{3})^{48}$ (corresponding to the $[0, 255]$ pixel range and $224 \times 224 \times 3$ extrinsic dimensionality of ImageNet images), we would then require about $(255 \times 224 \times \sqrt{3}/65)^{48} = 10^{153}$ points to generalize. Indeed, following the linear trend of Figure 3, we may expect to approximately halve the required inductive bias with this number of training points. However, with only $10^{17}$ points, strong inductive biases would be necessary to generalize.
> > > > >
> > > > > We have clarified these points in our revised Section 5 and have added further discussion in our revised Appendix G.

---

### Decision · Program_Chairs · 2023-01-20

**Decision:**

Reject

**Justification For Why Not Higher Score:**

As I said in my detailed meta-review, all the reviewers shared similar concerns about the clarity and readability of the paper. I think in its current form the paper won't be accessible to the reader and major revisions are required to improve its clarity.

**Justification For Why Not Lower Score:**

N/A

**Metareview: Summary, Strengths And Weaknesses:**

The paper claims to be the first to rigorously define a quantitative notion of the task complexity ("inductive bias complexity" of Definition 1). Roughly speaking, the authors propose to define the task difficulty through the fraction of predictors that generalize, among all the predictors that interpolate the training set. The definition is stated in a very broad, abstract, and general framework that captures multiple learning scenarios and tasks, popular int he machine learning field. The authors also provide theoretical upper bounds on the task complexity for some classes of problems, as well as empirical results on estimating the inductive bias complexity for some of the benchmark datasets.

All the reviewers (myself included) agree that the paper focuses on a very important direction that is, unfortunately, not very well covered by the current literature. Properly defining (hopefully in a computable way) the notion of task complexity would be very beneficial for machine learning community. Especially because this question naturally relates to the notion of the inductive bias of algorithms, which is known to play a central role in generalization.

Unfortunately, after reading the paper and based on the reviewer / rebuttal discussion, I have to conclude that in its current form the paper can not be accepted. There are two main reasons for this decision.

First, the reviewers (and myself) shared similar concerns related to the readability and *clarity* of the paper. The reviewers noticed that "in general the paper is hard to follow and unclear in some places, especially in Section 4.2". "Large parts of the paper are very informally written..., which makes it very difficult to assess the validity of the results". The paper "does not use notations consistently" and "arguments are often only vaguely outlined and prior work is used without giving much context". A larger list of places where the writing could be improved is provided by Reviewer NEm3.

Second, the reviewers feel that it is difficult to judge the *significance of the contributions*, largely due to the fact that the setup considered in the paper is too general. It was noticed by the reviewers that "while the great level of generality is certainly a strength of this work, it is also one of its bigger weaknesses. ... [the setup] is very unintuitive and not easy to follow" and also that some of the problems related to clarity "come from trying to accommodate many learning settings under one umbrella".  As a possible workaround the reviewers suggested that "maybe one just needs to consider only the standard supervised classification setting first." It was also mentioned that "the paper would benefit from providing a stronger motivation that highlights clearly how this measure of complexity [inductive bias complexity] can be useful to the community [by either] providing new insights or practice applications".

Finally, I would like to mention that the authors kept quiet, did not share the rebuttals and engage in active discussions until the *very last dat* of the paper revision and rebuttal discussion phase. Then the authors posted all the rebuttals, including a *very long* thread on the proposed paper revisions. Considering the fact that it happened right before the official deadline of the discussion phase, realistically the reviewers did not have a time to iterate on the rebuttals. I would like to encourage the authors to perform a major revision of their paper that accounts for everything discussed on this page and re-submit the result (which is likely very different from the version submitted to this conference) to one of the future conferences.

**Summary Of Ac-Reviewer Meeting:**

N/A